# Strong electron-phonon coupling driven pseudogap modulation and density-wave fluctuations in a correlated polar metal

Huaiyu (Hugo) Wang [1,7,8] ✉, Yihuang Xiong[1,2,8], Hari Padma[1], Yi Wang [1], Ziqi Wang[1], Romain Claes[3], Guillaume Brunin [4], Lujin Min [1], Rui Zu [1], Maxwell T. Wetherington[1], Yu Wang[5,6], Zhiqiang Mao [5,6], Geoffroy Hautier [2,3], Long-Qing Chen [1], Ismaila Dabo[1] ✉ & Venkatraman Gopalan [1] ✉

There is tremendous interest in employing collective excitations of the lattice, spin, charge, and orbitals to tune strongly correlated electronic phenomena. We report such an effect in a ruthenate, $Ca_3Ru_2O_7$, where two phonons with strong electron-phonon coupling modulate the electronic pseudogap as well as mediate charge and spin density wave fluctuations. Combining temperature-dependent Raman spectroscopy with density functional theory reveals two phonons, $B_2^P$ and $B_2^M$, that are strongly coupled to electrons and whose scattering intensities respectively dominate in the pseudogap versus the metallic phases. The $B_2^P$ squeezes the octahedra along the out of plane $c$-axis, while the $B_2^M$ elongates it, thus modulating the Ru 4d orbital splitting and the bandwidth of the in-plane electron hopping; Thus, $B_2^P$ opens the pseudogap, while $B_2^M$ closes it. Moreover, the $B_2$ phonons mediate incoherent charge and spin density wave fluctuations, as evidenced by changes in the background electronic Raman scattering that exhibit unique symmetry signatures. The polar order breaks inversion symmetry, enabling infrared activity of these phonons, paving the way for coherent light-driven control of electronic transport.

Mott Hubbard insulators are an example of strongly correlated electron system, where the energy scale for electron-electron repulsion is predominant relative to the kinetic energy of electrons[1,2]. On the other end, the energy scale for electron-phonon coupling (EPC) is predominant relative to the kinetic energy in a Peierls transition where charge density waves are observed[3]. In the intermediate regime of electronic correlation strength[4] between that of weakly correlated Fermi-liquid metals and strongly correlated Mott insulators, a rich

spectrum of competing electronic phenomena emerge such as unconventional superconductivity[5,6], pseudogap phase[7,8], and spin and orbital ordering[9]. In such systems, the energy scale for EPC can indeed be non-negligible in comparison with the correlation and exchange energy scales and could allow phonons to modulate emergent phenomena[10–12], such as possible signatures of superconductivity in cuprates[13–16], metal-insulator transitions in $TaS_2$[17], $VO_2$[18] and manganites[19–21]. Some of the ruthenates also lie in this intermediate

[1]Materials Research Institute and Department of Material Science & Engineering, Pennsylvania State University, University Park, PA 16802, USA. [2]Thayer School of Engineering, Dartmouth College, 14 Engineering Drive, Hanover, NH 03755, USA. [3]Institute of Condensed Matter and Nanosciences (IMCN), Université catholique de Louvain, Chemin des Étoiles 8, B–1348 Louvain-la-Neuve, Belgium. [4]Matgenix, Gozée, Belgium. [5]2D Crystal Consortium, Material Research Institute, Pennsylvania State University, University Park, PA 16802, USA. [6]Department of Physics, Pennsylvania State University, University Park, PA 16802, USA. [7]Present address: Stanford Institute for Materials and Energy Sciences, SLAC National Accelerator Laboratory, Menlo Park, CA 94025, USA. [8]These authors contributed equally: Huaiyu (Hugo) Wang, Yihuang Xiong. ✉e-mail: hugo17.wang@gmail.com; ixd4@psu.edu; vgopalan@psu.edu

correlation strength regime where correlation, exchange and EPC energy scales are all significant enough in influencing the physics underlying a rich spectrum of phenomena such as Mott insulators[22], superconductivity[23], pseudogap phase[24,25], magnetic[26], orbital[27] and polar orders[28]. $Ca_3Ru_2O_7$, the subject of this study exhibits an intermediate electron repulsion energy $U \sim 1$ eV, and onsite spin exchange coupling energy $J \sim 0.4$ eV[27,29,30]. However, despite two decades of research on this material, even the symmetry assignments of the phonons are currently incorrect; correcting these assignments in this study reveals new physics. Crucially, we show that the deformation potential (the square of which is proportional to the unitless EPC term $\lambda$[31]) for two of the correctly assigned $B_2$ phonons in this system can reach up to ~10 eV/Å, or ~0.1 eV per picometer of atomic displacement. Combining temperature dependent Raman spectroscopy with density functional theory (DFT), we show that these phonons (~52 meV) can modulate the electronic pseudogap (~10 meV) and mediate emergent density wave fluctuations that are absent in the ground state. The phonons are also infrared active, providing a pathway for potential optical control of the electronic transport instead of chemical pressure through doping as is conventionally explored[32].

$Ca_3Ru_2O_7$ is an $n = 2$ Ruddlesden-Popper phase of $Ca_{n+1}Ru_nO_{3n+1}$, situated at the phase boundary between a Fermi liquid system, $CaRuO_3$ ($n = \infty$), and a Mott insulator, $Ca_2RuO_4$ ($n = 1$). Slight impurity doping of 0.3% can induce a Mott insulating phase in $Ca_3Ru_2O_7$[33]. The ground state of $Ca_3Ru_2O_7$ is a perplexing pseudogap electronic phase whose physical origin is still debated. The pseudogap is a partial gap that opens in their spectral function at specific k points. Recent density functional theory (DFT) calculations predict that a charge density wave with alternating $RuO_6$ octahedral volumes can replicate the pseudogap band structure observed experimentally[34]. The presence of such density wave in the ground state has been postulated from the temperature dependent optical conductivity spectra[24] and ARPES[25] data. However, more recent ARPES[35,36] and diffraction[37–39] experiments have shown no evidence of density wave formation in the pseudogap phase. Although the ground state appears to lack the predicted density waves, we show that appropriate infrared optical phonons in $Ca_3Ru_2O_7$ can indeed induce the predicted density wave fluctuations.

There are two central findings in this work. Firstly, correcting the previous erroneous symmetry assignments of phonons[40] (Supplementary Note 1) reveals two dominant $B_2$ phonons, labeled $B_2^P$ and $B_2^M$, that strongly couple to the electronic and magnetic ground states: the scattering intensity of $B_2^P$ and $B_2^M$ are significantly enhanced respectively in the metallic and pseudogap phases. The

magnitude of the deformation potential of these two phonons in $Ca_3Ru_2O_7$ is comparable to known 2D systems with strong EPC, such as graphene[41] (14.1 eV/Å) and $MgB_2$[42] (6.5 eV/Å). First-principles frozen-phonon calculations with the important inclusion of spin-orbital coupling reveal that the $B_2^P$ phonon mode opens the pseudogap, while the $B_2^M$ mode closes it. Analysis of the eigen modes correlate the electronic pseudogap changes to phonon modulation of the Ru 4d orbital splitting and in-plane electron hopping. Secondly, a careful symmetry analysis of the background electronic Raman scattering reveals strong evidence for charge and spin density wave fluctuations mediated by the infrared active $B_2$ phonons. Below 48 K, the electronic Raman scattering shows symmetry-dependent spectral weight transfer manifested by a distinctive rise above 550 $cm^{-1}$ in the $B_2$ spectra which is absent in the $A_1$ spectra. This key symmetry signature in the electronic Raman scattering strongly points to the presence of charge and spin density wave like fluctuations mediated by optically excited $B_2$ phonons (~52 meV) below the metal-pseudogap transition at 48 K (4 meV). The charge and spin density wave phase has been predicted[34] in $Ca_3Ru_2O_7$ but has been experimentally found to be absent in the ground state[35–39]. Our findings deepen the understanding of the role of electron-phonon coupling in modulating the pseudogap in correlated materials and pave the way for potential phononic control of electronic phases in ruthenates.

## Results

### Large anomalous phonon renormalization across the spin-reorientation transition

The polar metal $Ca_3Ru_2O_7$ exhibits a rich range of ground states (Fig. 1b) realized by a subtle interplay between spin, charge, orbital and lattice degrees of freedom. $Ca_3Ru_2O_7$ is a paramagnetic (PM) metal above $T_N = 56$ K and is an antiferromagnetic metal with an easy axis along the $a$ axis between 48-56 K (AFM-$a$ phase, magnetic space group $Bb2_1'm'$). It then transforms into an antiferromagnet with its easy axis along $b$ axis below $T_C = 48$ K (AFM-$b$ phase, magnetic space group $Bb'2_1m'$); the structures of two AFM phases are shown in Fig. 1a. The structural refinement from neutron diffraction indicates that the polar 2-fold axis is along the longer in-plane $b$-axis ($b = 5.64$ Å)[43]; this is contrary to the assignment in a previous Raman study on this material[40], where the 2-fold axis was erroneously assigned to the shorter $a$-axis in-plane ($a = 5.43$ Å) (Supplementary Note 1). This error in the assignment of the 2-fold results in what are supposed to be the $B_2$ Raman modes as $A_2$ modes, and accordingly, the eigen displacements of these modes in the previous work[40] are incorrect.

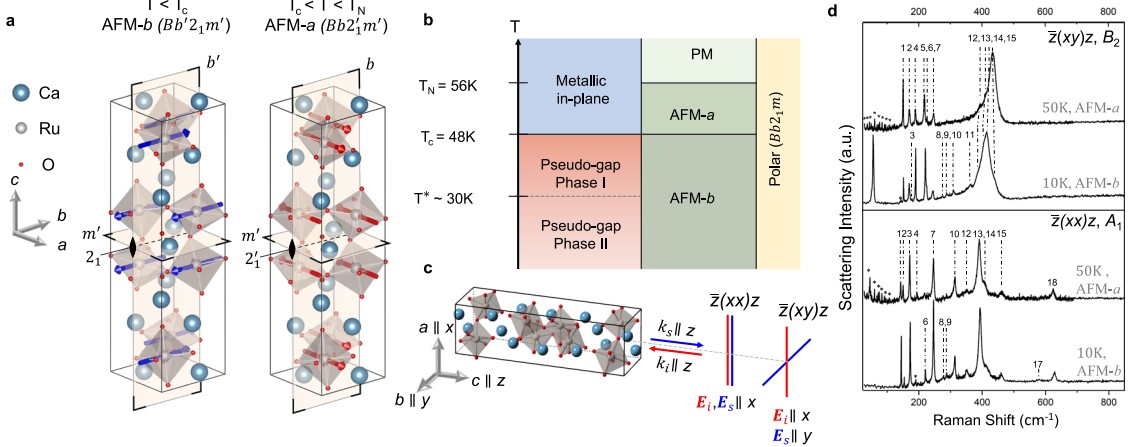

**Fig. 1 | Raman spectra coupled to the ground states of $Ca_3Ru_2O_7$. a** Crystal structure of $Ca_3Ru_2O_7$ with illustrated magnetic space group, symmetry operations and magnetic spins in the AFM-$a$ and AFM-$b$ phases. **b** Electronic and magnetic phase diagram of $Ca_3Ru_2O_7$. **c** Schematically depicted Raman lab geometry with respect to the crystal lattice of $Ca_3Ru_2O_7$. The $E_{i,s}$ and $k_{is}$ are polarization and wavevector or incident and scattered light respectively. **d** Representative polarized Raman intensities for $Ca_3Ru_2O_7$ at AFM-$b$ and AFM-$a$ phases respectively.

The experimental geometry (Fig. 1c) involves 632.8 nm light propagates parallel to the $c$ axis, while the light polarization is along the $a$ or the $b$ axes, and the lab coordinate system is defined such that $x \parallel a$, $y \parallel b$ and $z \parallel c$. There are two non-degenerate configurations that produce non-zero experimentally observable Raman signal: $\bar{z}(xx)z$ and $\bar{z}(xy)z$. The first and fourth letters in this notation respectively stand for the directions of propagation (wavevectors) of the incident ($i$) and scattered ($s$) light, whereas their polarizations are respectively denoted by the second and third letters. The character table in Table 1 correctly places the 2-fold symmetry parallel to the $b$ axis[44–46]. $\bar{z}(xx)z$ corresponds to the $A_1$ phonons and $\bar{z}(xy)z$ corresponds to the $B_2$ phonons. The polarized spectra of $Ca_3Ru_2O_7$ are measured at 50 K (AFM-$a$ phase) and at 10 K (AFM-$b$ phase) with 632.8 nm wavelength excitation (Fig. 1d). Similar spectra were observed in both the phases with 532 nm excitation (Supplementary Fig. 1), indicating that the Raman spectra presented are off-resonant. Self-consistent field calculations on electronic and phononic properties on $Ca_3Ru_2O_7$ are performed with DFT LDA with on-site Coulomb repulsion of 1.2 eV applied on the 4d orbitals and spin-orbit interaction included. We fit the experimental spectra in Fig. 1e, and the obtained phonon peak positions are consistent with those identified by first and second derivatives of the spectra (see Supplementary Fig. 4). The converged fitting results of experimental phonon energy agrees well with the DFT predicted

phonon energy (see Supplementary Note 3). The experimental fitting of phonon energy is robust against different background subtractions to account for the hybridization between phonon peaks and the electronic continuum (details in the Supplementary Note 4).

Reassignment of the $B_2$ phonons reveal a crucial difference from the previous literature[40,47]: the broad feature between 380 $cm^{-1}$ and 450 $cm^{-1}$ in the $B_2$ spectra (Fig. 2a) is composed of five phonon modes, indexed from $B_2^{(11)}$ to $B_2^{(15)}$, instead of one peak as identified in previous literature[48–50]. Further, the raw Raman spectra (Fig. 2b) shows that at 48 K near the phase transition, at least two peaks emerge as highlighted by the arrows. The experimental fitting of this composite peak is guided by frozen-phonon DFT calculations in the AFM-$a$ and AFM-$b$ phases. Comparison between the experimental and theoretical peak assignments (Fig. 2c, d) indicates a good agreement in both the AFM-$b$ and AFM-$a$ phases for these five phonons.

The assignment of five peaks within the $B_2$ Raman spectra from 350 to 450 $cm^{-1}$ is further examined due to the congested peak features within this energy range. We can identify the features of five peaks from the taking first and second derivatives of the raw data of $B_2$ Raman spectra as a function of temperature across $T_c$ (Supplementary Fig. 4). The presence of five peaks in the congested $B_2$ Raman spectra from 350 to 450 $cm^{-1}$ can also be indirectly inferred by 3% Ti doping of the crystal without changing the point group of the crystal, where the Raman results shows a splitting of peaks from $B_2^{(12)}$ to $B_2^{(15)}$ peaks into two pairs of peak features with clear shoulders (Supplementary Figs. 5 and 6 and Supplementary Table 4). (Ti-doped sample is not discussed further in this study, since it requires a separate follow-up study). We also carefully compare our fitting model of five peaks against using a single Fano-line shape suggested by previous literature[40] and conclude that our fitting model achieves better fits (Supplementary Fig. 7). For two phonon modes of interest, $B_2^{(13)}$ and $B_2^{(15)}$, the potential energy surface change is insensitive to the choice of DFT functionals (see Supplementary Fig. 8 in Supplementary Note 6). Thus, in this work, we

### Table 1 | Character table of *m2m* symmetry

| m2m | 1 | 2 | m(yz) | m(xy) | | |
|-----|---|---|-------|-------|---|---|
| $A_1$ | 1 | 1 | 1 | 1 | y | $x^2$, $y^2$, $z^2$ |
| $A_2$ | 1 | 1 | −1 | −1 | $R_y$ | xz |
| $B_1$ | 1 | −1 | 1 | −1 | z, $R_x$ | yz |
| $B_2$ | 1 | −1 | −1 | 1 | x, $R_z$ | xy |

Raman polarizability symmetry of $\bar{z}(xx)z$ and $\bar{z}(xy)z$ corresponds to $A_1$ and $B_2$.

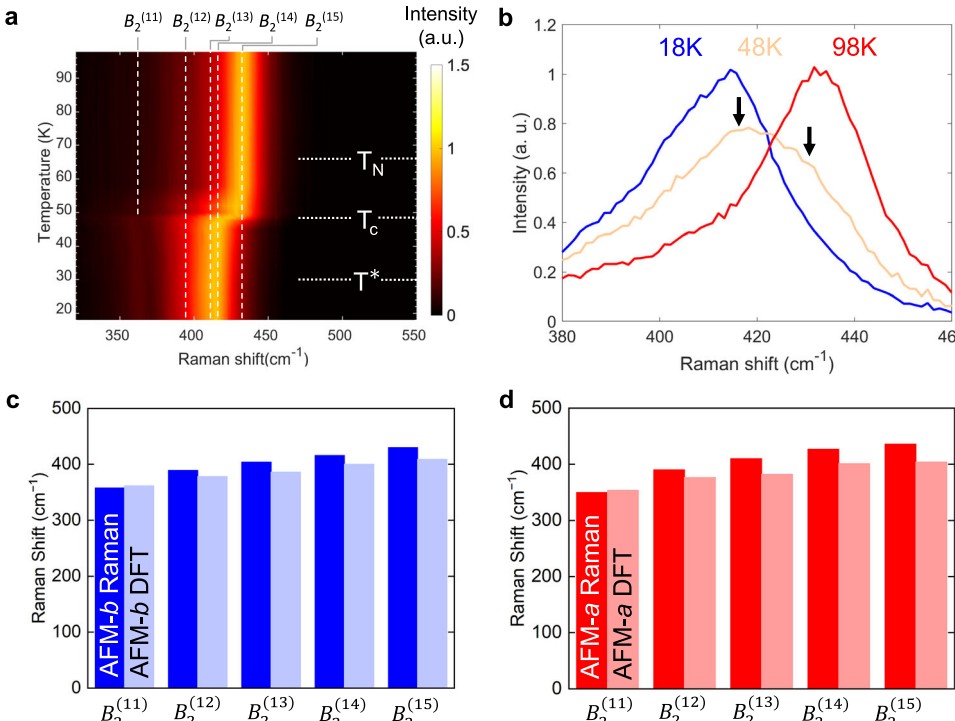

**Fig. 2 | Temperature dependent Raman spectra and a comparison with density functional theory. a** Temperature dependent Raman spectra in $\bar{z}(xy)z$ with $B_2^{(11)}$ to $B_2^{(15)}$ modes highlighted. The intensity is normalized to strongest Raman peak. **b** $B_2$ spectrum at 48 K ($T_c$) showcase features of shoulder peaks highlighted with black arrows, compared with spectra at 18 K and 98 K. The DFT calculated $B_2^{(11)}$ to $B_2^{(15)}$ phonon energies compared to experimental observations in **c** AFM-$a$ and **d** AFM-$b$ phases.

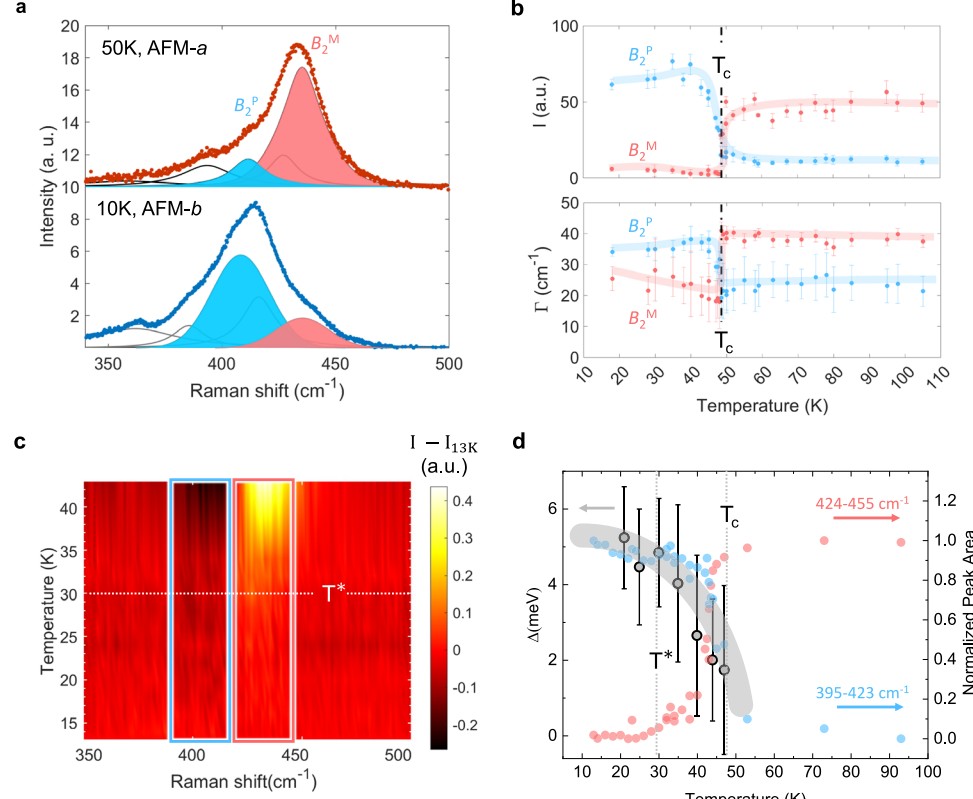

**Fig. 3 | Temperature-dependent reconstruction of $B_2^M$ and $B_2^P$ phonon spectra.**
**a** Fitting of $\bar{z}(xy)z$ Raman spectra between 340 cm$^{-1}$ and 470 cm$^{-1}$ at 50 K (top panel) and 10 K (bottom panel). the celeste/pink color-coded peak correspond to $B_2^P/B_2^M$.
**b** The temperature dependent plots of amplitude ($I$) and FWHM ($\Gamma$) of $B_2^P$ (celeste) and $B_2^M$ (pink) modes. Error bars are standard deviations from fitting values. The celeste/pink lines are guides to the eye. **c** Raman colormap plot of the intensity difference between specified temperature and 13 K. The total area of region of interest highlighted in blue and red boxes in **c** are replotted in **d** to indicate an intensity anomaly observed at T* = 30 K. The pseudogap size ($\Delta$) is extracted from ARPES data[35] and is represented as grey data points. The broad grey line is a guide to the eye.

present both experimental and computational evidence that the broad feature is composed of five phonon modes (instead of one) and are indexed from $B_2^{(11)}$ to $B_2^{(15)}$.

We present the region of interest in the $B_2$ phonon spectra at two representative temperatures in Fig. 3a where clearly the $B_2^{(13)}$ (henceforth labeled $B_2^P$) mode amplitude is suppressed while the $B_2^{(15)}$ (henceforth labeled $B_2^M$) mode amplitude is enhanced upon heating above $T_c$ (For fitting results of all observed modes, see Supplementary Figs. 9–12 in Supplementary Note 7). The overall temperature dependent fitting results (Fig. 3b) of $B_2^P$ and $B_2^M$ show a clear crossover of the Raman scattering intensity as well as that of the full-width-at-half-maximum (FWHM).

To understand these experimental trends, let us consider the origin of the Raman scattered intensity including EPC, which is given as follows[51]:

$$I = \alpha^2 F(\omega) \approx \frac{2}{\pi} N(0) \omega \sum_\nu \Gamma_{\mathbf{q}=\mathbf{0},\nu}^{ep} \delta\left(\omega - \omega_{\mathbf{q}=\mathbf{0},\nu}\right) \quad (1)$$

where $\alpha$ is EPC term; $F(\omega)$ is the phonon density of state at phonon frequency $\omega$; $N(0)$ is the electronic density of states at the Fermi surface, $\Gamma_{\mathbf{q},\nu}^{ep}$ is the electron-phonon-induced Raman peak width, $\omega_{q,\nu}$ is the phonon frequency of index $\nu$ and wavevector $\mathbf{q}$. The Raman linewidth, $\Gamma_{\mathbf{q},\nu}^{ep}$, is given by the Fermi golden rule[51,52] as follows:

$$\Gamma_{\mathbf{q},\nu}^{ep} = 2\pi \sum_{\mathbf{k},i,j} \left| g_{(\mathbf{k}+\mathbf{q})j,\mathbf{k}i,\nu}^{ep} \right|^2 \left[f_{\mathbf{k}i} - f_{(\mathbf{k}+\mathbf{q})j}\right] \times \delta\left[\epsilon_{\mathbf{k}i} - \epsilon_{(\mathbf{k}+\mathbf{q})j} + \hbar\omega_{\mathbf{q},\nu}\right] \quad (2)$$

Here, $f$ is the Fermi-Dirac distribution function; $\epsilon$ is the energy of the electron; $g_{(\mathbf{k}+\mathbf{q})j,\mathbf{k}i,\nu}^{ep}$ is the EPC matrix for scattering from an electron state $\mathbf{k}$ located in band indexed $i$ to $\mathbf{k}+\mathbf{q}$ in band indexed $j$, via a phonon of wavevector $\mathbf{q}$ and mode index $\nu$.

From the above equations, one can conclude that both the Raman intensity, $I$, and the FWHM are proportional to the determinant of the EPC matrix, $|g_{(\mathbf{k}+\mathbf{q})j,\mathbf{k}i,\nu}^{ep}|$, given that the joint DOS is not significantly modulated by the phonon perturbation (Supplementary Fig. 13 in Supplementary Note 8). An estimate for the deformation potential, which indicates EPC strength, can be obtained from $\Gamma_{\mathbf{q}=\mathbf{0},\nu}^{ep}$ by excluding the combined contributions of the phonon-phonon scattering and the inhomogeneous broadening which is estimated to be ~6 $cm^{-1}$ at $T_c$ from lower energy $B_2$ phonons (Supplementary Fig. 9). Due to the quasi-2D nature of the ground state[28,53], Ca$_3$Ru$_2$O$_7$ exhibits a linear band dispersion near the Fermi level[25,35], which justifies applying the 2D model[41,52] based on the Fermi golden rule: $\Gamma_{\mathbf{q}=\mathbf{0},\nu}^{ep} = \frac{A_{uc}}{8Mv^2}D^2$, where $A_{uc}$ (=5.365Å × 5.535Å) is the area of the quasi-2D unit cell[43] of Ca$_3$Ru$_2$O$_7$, $M$ (=434.374 g/6.02×10$^{23}$) is the unit cell mass[54], $v_F = 5 \times 10^4$ m/s is the Fermi velocity[25], and $D$ is the deformation potential, which is related to EPC matrix by $g_{(\mathbf{k}+\mathbf{q})j,\mathbf{k}i,\nu}^{ep} = D_{(\mathbf{k}+\mathbf{q})j,\mathbf{k}i,\nu} \sqrt{\hbar/(2M\omega_{\mathbf{q},\nu})}$. The measured $\Gamma_{\mathbf{q}=\mathbf{0},\nu}^{ep}$ of $B_2^P$ shows an increase of the estimated deformation potential from $D \sim 6.7$ eV/Å in AFM-$a$ phase to $D \sim 9.7$ eV/Å in AFM-$b$ phase. The $B_2^M$ shows a decrease of the estimated deformation potential from $D \sim 10.0$ eV/Å in AFM-$a$ phase to $D \sim 5.9$ eV/Å in AFM-$b$ phase.

To further elaborate on the strong EPC of the two dominant $B_2$ phonons, we zoom into the AFM-$b$ phase where the pseudogap size

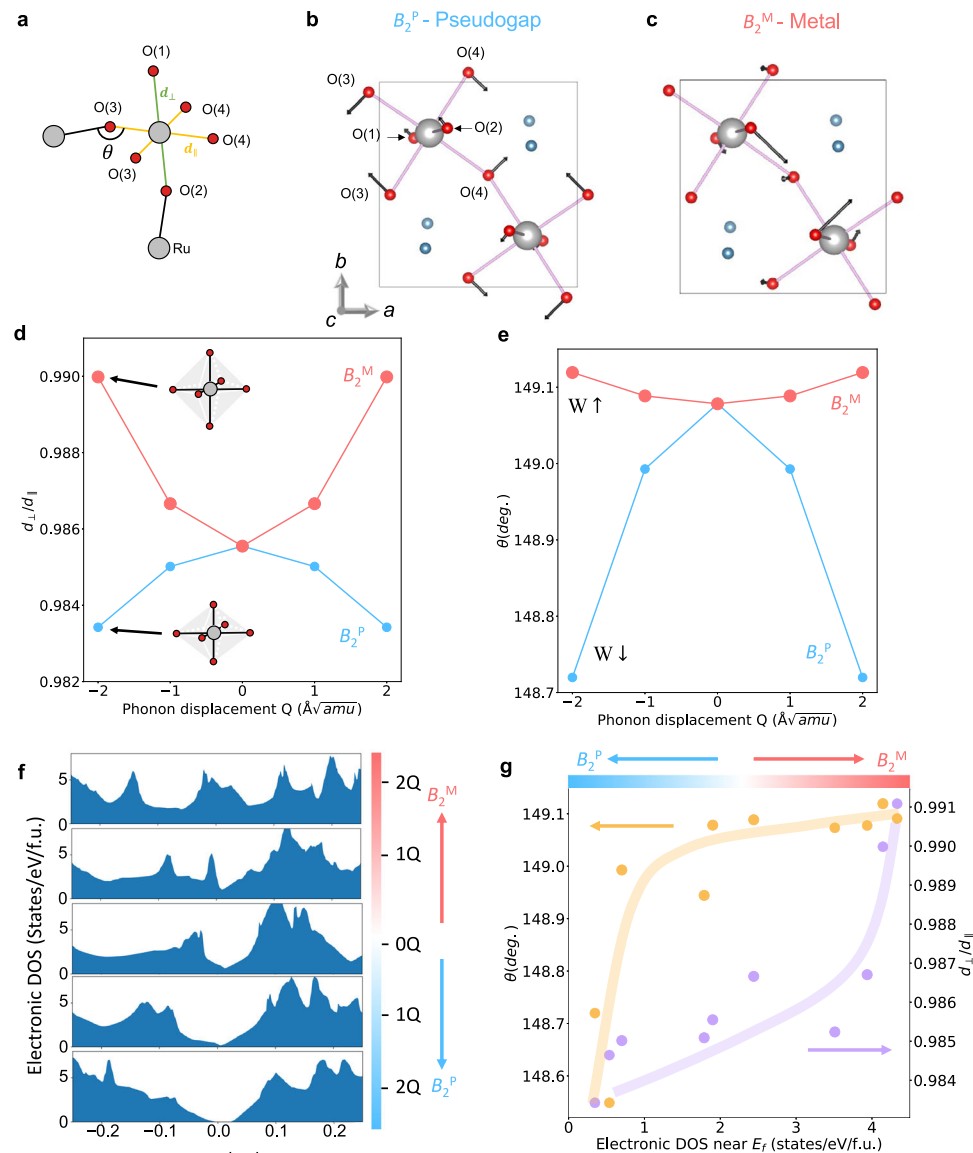

**Fig. 4 | Phonon eigenmodes and microscopic mechanism of electronic band modulation. a** Illustration of the Ru and O atoms, bond lengths between oxygens and the bond angle of O-Ru-O. **b** The illustrations of the eigen mode of $B_2^P$, and **c** that of $B_2^M$. **d** The averaged ratio of RuO$_6$ octahedra cage apical Ru-O bond length ($d_\perp$) over in-plane bond length ($d_\parallel$) modulated by two phonons. **e** Averaged in-plane bond angle ($\theta$) modulated by the two phonons. (see Supplementary Figs. 15 for phonon modulated bond length and angles in two types of RuO$_6$) **f** The density of states plots of the AFM-$b$ ground state (middle) and the modulated AFM-$b$ state by $B_2^M$ (top) and $B_2^P$ (bottom). **g** Correlation between octahedra distortion ($d_\perp/d_\parallel$), in plane bond angle ($\theta$) and the density of states near the Fermi surface. The lines are guides to the eye.

gradually increases as the temperature decreases. As shown in Fig. 3c, the Raman intensity changes ($I$ - $I_{13K}$) exhibit two distinct regions, one where the change increases and the other where it decreases as the temperature decreases. These two regions are chosen according to the peak positions of the $B_2^P$ (celeste) and $B_2^M$ (pink) phonons due to their large Raman intensity changes across T$_c$. The area in these regions is integrated to represent the two respective phonons; the intensity of nearby phonon modes, $B_2^{(12)}$ and $B_2^{(14)}$, do not change significantly (Supplementary Fig. 10). The pseudogap size correlates well with Raman intensity as a function of temperature across the metal-to-pseudogap phase transition (Fig. 3d). Further, the Raman intensity is close to saturation at T$^*$ but there is no abrupt anomaly in the phonons at T$^*$ indicating that T$^*$ is not a distinct phase transition; this is consistent with the experimental evidence seen in the recent optical pump probe study[35]. On the other hand, T$^*$ likely corresponds to an

anisotropic change in the electronic band structure near the Fermi-Surface[35].

## Microscopic Mechanism for the coupling of the two $B_2$ phonons to the Pseudogap

We investigate the coupling of the $B_2^P$ and $B_2^M$ phonons to the electronic structure via the frozen-phonon approach in DFT. The eigenmode of $B_2^P$ in the AFM-$b$ phase and that of the $B_2^M$ in the AFM-$a$ phase are shown in Fig. 4a–c. The $B_2^P$ mainly modulates in-plane Ru-O bonding lengths, $d_\parallel$, as a "scissoring" mode in which the two nearest oxygen atoms move toward and away from each other. On the other hand, the $B_2^M$ mainly modulates the apical Ru-O bonding lengths, $d_\perp$, between two perovskite layers (for numerical phonon eigenvectors of $B_2^M$ and $B_2^P$ and further analysis of the two modes, see Supplementary Tables 5, 6 and Supplementary Figs. 14–16 in Supplementary Note 9).

In $Ca_3Ru_2O_7$, the octahedral cage without any phonon distortions is slightly compressed in its equilibrium phase[43], since on the average, $\frac{d_\perp}{d_\parallel} < 1$. The $B_2^P$ modulation further compresses the $RuO_6$ octahedron along $d_\perp$ direction (Fig. 4d), while the $B_2^M$ mode elongates it irrespective of the sign of the Eigen vectors. Such octahedral distortions are linked to the occupation of the Fermi level DOS in strongly correlated systems[32]. Due to the quasi-2D nature of the ground state[28,53], only the in-plane Ru-O hopping parameter affects the electronic ground state. The averaged in-plane Ru-O-Ru bond angle $\theta$ plotted in Fig. 4e is largely suppressed by the $B_2^P$ mode, while the angle is slightly increased by the $B_2^M$ mode. The decreasing bond angle leads to less electron cloud overlap between Ru 4d and O 2p orbitals which reduces the electronic bandwidth W. The DFT calculation results in Fig. 4f shows that the pseudogap opens/closes as the ground state is modulated by the $B_2^P/B_2^M$ phonons, respectively. A clear correlation is seen (Fig. 4g) between the $\frac{d_\perp}{d_\parallel}$, $\theta$ and N(0) from the DFT calculations using the frozen phonon modulated DOS: $B_2^P/B_2^M$ phonons respectively decrease/increase the $\frac{d_\perp}{d_\parallel}$, the $\theta$, and hence the N(0) in the metallic and pseudogap phase. Although the observed phonons are not thermally excited to induce the pseudogap opening at $T_c$, the modulation of the pseudogap by $B_2^M$ and $B_2^P$ phonons revealed by first principles calculations highlights the key role of structure distortion in the Mott transitions which is in agreement with the structural changes underlying the insulating phases in Ti-doped $Ca_3Ru_2O_7$[56,57] and $Ca_2RuO_4$[58].

## Density wave fluctuations from phonon-assisted structural distortions

Next, we present evidence for density wave fluctuations from the Raman electronic background. The imaginary part of the electronic Raman response function ($\chi_u''(\omega, T)$) of $Ca_3Ru_2O_7$ is obtained after subtracting the phonon and magnon peaks to get electronic Raman scattering intensity ($I_u(\omega, T)$) and correcting for the Bose factor ($n$): $\chi_u''(\omega, T) \propto \frac{1}{(1+n(\omega,T))} I_u(\omega, T)$. In turn, $\chi_u''(\omega, T)$ probes the weighted charge correlation function: $\chi_u(\omega, T) = \langle \rho_u(\omega)\rho_u(-\omega) \rangle$, where $\rho_u(\omega) = \sum_k \gamma_u(k)N(k)$ is related to charge density operator $N(k)$ and Raman vertex $\gamma_u(k)$, which denotes the scattering amplitude governed by the Raman selection rule (Supplementary Note 10). Thus, Raman response function has sensitivity to electron transitions in reciprocal space where $|\gamma_u(k)| \neq 0$.

The experimental Raman response function $\chi_u''(\omega, T)$ as well as the Raman vertex, $\gamma_u(k)$, of the $B_2$ ($\gamma_{xy}$) and the $A_1$ (linear combination of $\gamma_{x^2+y^2}$ and $\gamma_{x^2-y^2}$) phonons computed from group theory are depicted in Fig. 5a and c, respectively. The Fermi surface information[25,35,36] combined with $\gamma_u(k)$ (insets of Fig. 5a, c) indicates that the $B_2$ spectra are mostly sensitive to the pseudogap feature centered around ($\pm\frac{\pi}{2a}$, $\pm\frac{\pi}{2a}$), and the $A_1$ spectra are mostly sensitive to metallic response at the $M$ and $M_I$ points, and are less sensitive (albeit with non-zero response) to the pseudogap features around the $\Gamma$ and the quasi-1D bands close to $M$ and $M_I$. Indeed, the pseudogap features shows up as a dip in differential Raman response $\Delta\chi_u''(\omega, T) = \Delta\chi_u''(\omega, T) - \Delta\chi_u''(\omega, 49K)$ below ~400 cm$^{-1}$ in both $B_2$ and $A_1$ spectra (Fig. 5b, d). From Gaussian fitting of the pseudogap dip feature, the probed pseudogap size is $2\Delta_{pseudogap} = 22.5 \pm 1.4 meV$ from $B_2$ Raman response function and $2\Delta_{pseudogap} = 21.4 \pm 1.3 meV$ from the $A_1$ Raman function at 28 K. This agrees well with the average pseudogap size extracted from optical conductivity measurements[24], which was reported to be ~20 meV. By integrating the dip area between 120 cm$^{-1}$ and 400 cm$^{-1}$ in both the $B_2$ and the $A_1$, the differential Raman responses avoid the Drude response at lower frequency; with this analysis, we observe that the area of the dip feature can be well described by a BCS type gap ($\Delta(T) \propto \tanh(2.2\sqrt{T_c/T - 1})$), which is consistent with ultrafast optical study on $Ca_3Ru_2O_7$[55] (Fig. 5e). Thus, the electronic Raman scattering response is a sensitive probe of the changes in the pseudogap.

In addition to the pseudogap feature, a hump in the differential Raman response is seen in $\Delta\chi_{B_2}''(\omega, T)$ above ~550 cm$^{-1}$ but is conspicuously absent in $\Delta\chi_{A_1}''(\omega, T)$ (Fig. 5b, d). In literature, we can an find excellent agreement of this hump feature with the optical conductivity spectra[24]. The feature was assigned to electron-hole quasiparticle excitation across the neighboring Ru atoms from $d_{xy}$ to $d_{xz/yz}$ that benefits from the energy gain through Hund's coupling (Supplementary Note 11). However, the absence of the hump features in $\Delta\chi_{A_1}''(\omega, T)$ in our present study is rather surprising and suggests new insights into its origin.

Both $\chi_{A_1}''(\omega, T)$ and $\chi_{B_2}''(\omega, T)$ have sensitivity to $d_{xz/yz}$ quasi-1D band near the Fermi-surface[25,35] where proposed Fermi nesting feature is at. The pseudogap dip area in $\Delta\chi_{B_2}''(\omega, T)$ is not sensitive to the laser line intensity, while the increasing intensity of incident laser gives rise to an increase in the $B_2^P$ phonon amplitude below $T_c$ (Supplementary Fig. 17). Furthermore, the intersite quasiparticle excitation hump area, $\Delta\chi_{B_2}''(\omega, T)$, above ~550 cm$^{-1}$ is linearly proportional to the $B_2^P$ phonons amplitude (Fig. 5e). It is highly likely that since $B_2^P$ mode shows an anomalously large EPC (order of ~10 eV/Å) below $T_c$, which is not observed in the FWHM of other $B_2$ modes (Supplementary Figs. 9 and 10), the intersite excitation appears to be mainly mediated by the $B_2^P$ mode. However, it is worth noting that Raman cannot uniquely single out the $B_2^P$ mode (Supplementary Fig. 18). A further look into the selection rule for intersite $d_{xy}$ to $d_{xz/yz}$ orbital electron-hole quasiparticle excitation suggests that in-plane excitation of electrons from $d_{xy}$ to O $2p_z$ is forbidden via the electric dipole transition between electronic states, unless the excitation is mediated by the $B_2$ phonons between vibronic states.

Using group theory analysis (Supplementary Note 13), we propose a 3-step mechanism (Fig. 5g) to explain the electronic background above ~550 cm$^{-1}$ using symmetry resolved quasiparticle information from the electronic Raman. The electron hopping is initiated in Step I by two simultaneous steps: (1) charge excitation from O $p_z$ orbital to Ru $d_{xz/yz}$ orbital (octahedra 1) and (2) the electron from neighboring Ru $d_{xy}$ (octahedra 2) hopping to the O $p_z$ orbital, assisted by the $B_2$ phonons. Then in Step II, the charge imbalance between the two $RuO_6$ cages would promote the electron hopping from (1) O $p_z$ to Ru $d_{xz/yz}$ (octahedra 2) and followed by electron hopping from (2) Ru $d_{xz/yz}$ (octahedra 1) to O $p_z$. Eventually, the electron configuration would return to the ground state (Step III). This provides a unique case where symmetry-breaking phonons drive charge and spin fluctuation like a density wave modulation by exciting electrons across neighboring $RuO_6$. The energy cost for this hopping is estimated as following: Hund's coupling energy benefits the intersite charge transfer and the energy cost is $U - 3J + D - \omega_{B_2}$ (Supplementary Note 11), where $U$ is the electron correlation energy, $J$ is the Hund's coupling energy, $D$ is the gap splitting between $d_{xy}$ and $d_{xz/yz}$ and $\omega_{B_2}$ is the gain of phonon energy via vibronic mixing. With reported values of $U \sim 1$ eV, $J \sim 0.4$ eV, $D \sim 0.3$ eV[27,29,30] and considering the phonon energy of $B_2^P$ (0.05 eV), this would result in an energy barrier of ~0.05 eV for the $d^4 + d^4 \rightarrow d^3 + d^5$ electron transition; this agrees with our experimental result that shows an increase in the electronic Raman scattering above 550 cm$^{-1}$ ~ 0.07 eV.

Based on this clear symmetry-based evidence for the density wave fluctuations mediated by the $B_2$ phonons (but not the $A_1$ phonons), a revisiting of the ground state of $Ca_3Ru_2O_7$ is merited. The formation of static charge density waves requires structural distortion. With the most recent high energy X-ray pair distribution function study on $Ca_3Ru_2O_7$[59], which has larger precision on Ru-O bond length than required to resolve the proposed static density wave phase, the space group remains $Bb2_1m$, which indicates that the static density wave distortion is not observed in the low temperature, in which case the predicted space group would change[34]. Below the first order transition

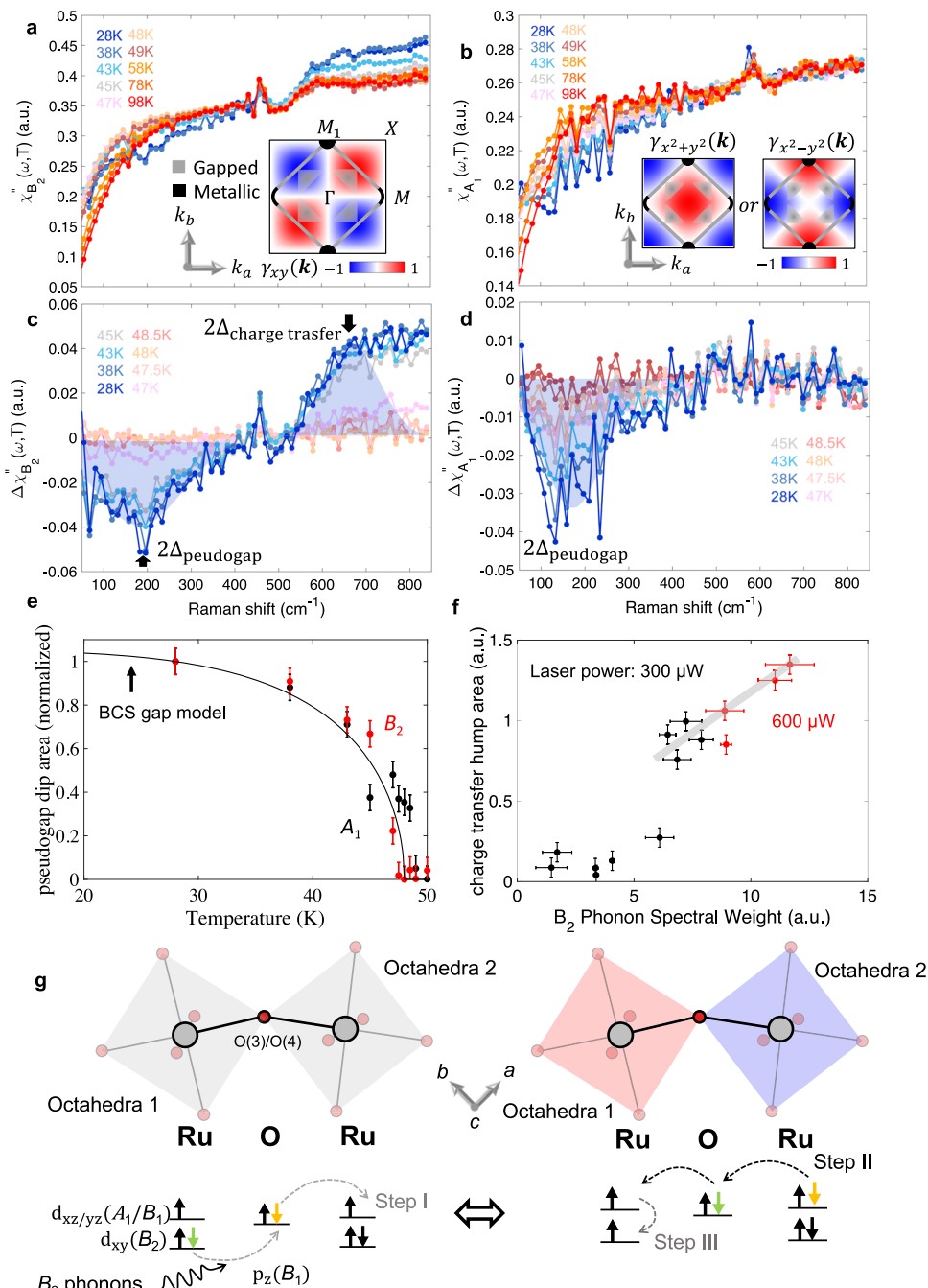

**Fig. 5 | Phonon mediated charge density fluctuation observed in electronic Raman scattering.** Temperature dependence of **a** $B_2$ and **b** $A_1$ Raman response function ($\chi''_u(\omega,T)$) of $Ca_3Ru_2O_7$ across $T_c$. The corresponding momentum-space structure of the form factor $\gamma^\mu_k$ are depicted in the insets (Supplementary Note 10) overlaid with a schematic of momentum resolved electronic band features in the low-energy excitation near the Fermi surface from ARPES studies[25, 35, 36]. The background responses of **c** $B_2$ and **d** $A_1$, after subtracting the background response at 49 K to highlight the dip feature below 400 cm$^{-1}$ arising from the pseudogap opening, and the hump feature seen above 550 cm$^{-1}$. **e** Temperature dependence of the pseudogap Raman signal obtained by summing between 120 cm$^{-1}$ and 400 cm$^{-1}$

in $\Delta\chi''_{B_2}(\omega,T)$ (red circles) and $\Delta\chi''_{A_1}(\omega,T)$ (black circles). The signal is normalized to the maximum signal at 28 K and the solid line is a BCS gap model. **f** Correlation between $B_2^P$ spectral weight and charge transfer Raman response at $300\mu W$ (black circles) and $600\mu W$ (red circles) incident laser power. **g** A schematic diagram of the proposed 3 step mechanism for the hump feature in $B_2$: the orbital flip excitation is mediated by $B_2$ phonons. The orbital flip excitation transiently differentiates the neighboring $RuO_6$ octahedra and induces dynamical charge and spin density fluctuations. Errorbars of integrated area is estimated from variance of featureless section in $\Delta\chi''_u(\omega,T)$, and the errorbars of phonon spectral weight are standard deviations from fitting results.

temperature $T_c = 48$ K, the bond angles and bond length below $T_c$ undergo a continuous transition as a function of temperature but the structure distortion does not promote electron hopping between intersite Ru atoms[39,43] (see detailed analysis in Supplementary Note 14). This indicates that synchronized modulation of the in-plane electron hopping integral by a phonon such as $B_2^P$ is a viable means to drive the

structure towards a charge density wave distortion (see Supplementary Figs. 16a, 20). Indeed, temperature dependent electronic Raman suggests that the $B_2^P$ phonon assists in the formation of the density wave fluctuations in $Ca_3Ru_2O_7$. While the density wave fluctuation is largely incoherent and short-lived as indicated by the broad nature of the hump feature (for the discussion of the lifetime of the density wave

fluctuation, see Supplementary Note 15), resonantly driving the infrared (IR) active $B_2^P$ mode with mid IR pulses is a promising avenue to create long-lived charge order states in $Ca_3Ru_2O_7$.

Using Raman spectroscopy combined with DFT calculations, this work presents a significant revision of the symmetry assignment of previously reported phonon spectra[40] in $Ca_3Ru_2O_7$, which alters the interpretation of underlying physics. In particular, two phonon modes labeled $B_2^P$ and $B_2^M$ show a strong cross-over effect in their mode specific deformation potential across the metal-pseudogap transition temperature $T_c = 48$ K. Employing DFT calculations with spin-obit coupling, we discover the microscopic mechanism through which the two phonon modes close and open the pseudogap, respectively. Two phonons modulate the bandwidth near the Fermi surface and the $RuO_6$ octahedral compression ratio, which in turn are coupled to the pseudogap modulation. Temperature-dependent electronic Raman scattering presents evidence for the formation of the phonon-mediated charge and spin density fluctuation that has been predicted by theory but absent from ground state[34]. Our study indicates that this density wave phase can be potentially stabilized by resonant light excitation of IR active $B_2^P$ phonon mode. Our finding in $Ca_3Ru_2O_7$ motivates a potential pathway for modulating competing phases by phonon excitations using light in correlated systems with strong electron-phonon coupling.

## Methods

### Sample preparation

High quality calcium ruthenate $Ca_3Ru_2O_7$ single crystals were grown by the floating zone techniques using a mirror-focused furnace[60], and aligned ex situ by X-ray Laue diffraction for polarized Raman measurement. The composition of the sample was determined via X-ray diffraction.

### Raman spectroscopy measurements

The Raman experiments were carried out in a 4 K helium-flow Oxford microstat He cryostat connected to an Agilent TPS-compact pump station with base pressures of $5 \times 10^{-6}$ torr. Temperature-dependent Raman spectra were collected using a Horiba LabRam HR Evolution with a freespace Olympus BX51 confocal microscope. A 632.8 nm linearly polarized HeNe laser beam was focused at normal incidence using a LWD 50x objective with a numerical aperture of 0.5, with the confocal hole set to 100 $\mu$m. A Si back-illuminated deep depleted array detector and an ultra-low-frequency volume Bragg filter were used to collect the spectra, dispersed by a grating (1800 gr/mm or 600 gr/mm) with an 800 mm focal length spectrometer.

The laser power was maintained below 300 $\mu$W in all the temperature-dependent measurements using 600 gr/mm (unless stated otherwise), to minimize laser heating and maintain the power well below the damage threshold. The 1800 gr/mm data was collected using 600 $\mu$W 632.8 nm laser and 400 $\mu$W 532 nm laser. The temperature presented in this paper is after correcting for the laser heating. Laser heating was calibrated as described in the Supplementary Note 16 with thermal conductivity values from reference[55]. Polarized spectra were obtained using a half-waveplate to rotate the polarization of the incident beam, with a fixed analyzer. Spectra were averaged over 60 minutes in the case of temperature-dependent measurements, with a temperature stability of $\pm 0.1$ K. The temperature dependent spectra are shown in Supplementary Fig. 2. The intensity of colormap in Supplementary Fig. 2 is normalized to the Raleigh line intensity.

### Raman peak fitting

The $A_1$, and $B_2$ peaks were fit with a standard Lorentz lineshape, except the $B_2^{(13)}$ and $B_2^{(15)}$ peaks that were fit with a Pseudo Voigt lineshape. A nonlinear least-squares fitting procedure was used. To ensure robustness of the temperature-dependent fits, the same initial fit values and

constraints were used for each set of temperature-dependent spectra. The $B_2$ peaks were fit after subtracting the background with Savitzky-Golay filter to account for potential hybridization between background and phonon peaks. The experimental fitting is robust against different background subtractions. Detailed description of background subtraction method and robustness of fitting demonstration can be found in the Supplementary Note 4 and 5.

### Electronic structure and phonon calculations

Self-consistent field calculations are performed with the Vienna Ab-initio Simulation Package (VASP) using projected augmented wave method[61,62]. To correct for the self-interaction issue, the on-site Coulomb repulsion of 1.2 eV is applied on the 4d orbitals using Dudarev's approach[34,55,63]. The spin-orbit interaction is included for the atomic position optimizations. The lattice parameters in all the DFT studies are from the reported experimental structures with a space group of $Bb2_1m$[43]. Our results show that AFM-$b$ is more stable than AFM-$a$ (with an energy difference of 17.3 meV). For low and high-temperature phases, the bi-layer antiferromagnetic orderings are respectively along the [001] and [010] axes, which correspond respectively to the AFM-$a$ and AFM-$b$ phases. Previous literature[64] has shown that LDA performs better than PBEsol for describing the phononic properties in $Ca_3Ru_2O_7$. Therefore, LDA functional[65] is utilized to study the electronic and phononic properties in this work. The optimization of the atomic positions is carried out with a plane-wave cutoff of 650 eV and the Brillouin zone is sampled using a Gaussian smearing with a 20 meV width on a $10 \times 10 \times 6$ $\Gamma$-centered $k$-mesh. The energy and forces are converged to be within $10^{-8}$ eV and 0.1 meV/Å. The phononic properties are calculated using frozen phonon approach that is implemented in phonopy[66]. We represent phonons using mass-weighted atomic displacements $Q_{I\alpha} \equiv \sqrt{m_I} u_{I\alpha}$, where $I$ is the atom index, $\alpha$ is the space direction, $m_I$ is the mass of atom $I$ (in atomic units of mass, a.m.u = $1.6605 \cdot 10^{-24}$ g), and $u_{I\alpha}$ is the displacement of atom $I$ along direction $\alpha$ (in Å). The dynamical matrix $\boldsymbol{D}(\boldsymbol{q})$ (the mass-rescaled interatomic force constant matrix for monochromatic perturbations of atomic positions of wave vector $\boldsymbol{q}$) can then be expressed as

$$D_{I\alpha,J\beta}(\boldsymbol{q}) = \frac{1}{\sqrt{m_I m_J}} \sum_{\boldsymbol{R}} e^{i\boldsymbol{q} \cdot \boldsymbol{R}} \Phi_{I\alpha,J\beta}(\boldsymbol{R}),$$

where $\Phi_{I\alpha,J\beta}(\boldsymbol{R}) = \partial^2 E / \partial u_{I\alpha}(\boldsymbol{0}) \partial u_{J\beta}(\boldsymbol{R})$ is the (un-rescaled) interatomic force constant matrix between atoms $I\alpha$ and $J\beta$ residing in unit cells that are separated by the translation vector $\boldsymbol{R}$. Since the Raman response is related to the zone-center phonons ($\boldsymbol{q} \approx \boldsymbol{0}$), we consider the dynamical matrix at the $\Gamma$-point:

$$D_{I\alpha,J\beta}(\boldsymbol{0}) = \frac{1}{\sqrt{m_I m_J}} \sum_{\boldsymbol{R}} \frac{\partial^2 E}{\partial u_{I\alpha}(\boldsymbol{0}) \partial u_{J\beta}(\boldsymbol{R})} = \sum_{\boldsymbol{R}} \frac{\partial^2 E}{\partial Q_{I\alpha}(\boldsymbol{0}) \partial Q_{J\beta}(\boldsymbol{R})}.$$

By diagonalizing the dynamical matrix, one obtains the resonant frequencies $\omega_n$ and the normal modes $\boldsymbol{e}_n$:

$$\sum_{j\beta} D_{I\alpha,J\beta}(\boldsymbol{0}) e_{J\beta,n} = \omega_n^2 e_{I\alpha,n}.$$

Therefore, the atomic displacements can be decomposed along the normal modes as

$$u_{I\alpha} = \frac{Q_{I\alpha}}{\sqrt{m_I}} = \frac{1}{\sqrt{m_I}} \sum_n Q_n e_{I\alpha,n},$$

where $Q_n \equiv \sum_{I\alpha} Q_{I\alpha}^* e_{I\alpha,n}$ denotes the coordinate of the $n^{th}$ normal mode [in $(a.m.u)^{1/2}$ Å].

The electronic band structures at $k_z = 0$ for the AFM-$a$ and AFM-$b$ phases are shown in Supplementary Fig. 22. LDA + $U$ + SOI predicts

a suppressed band crossing of the low-temperature AFM-*b* compared to the AFM-*a* phase with a prominent effect near $M_1$, which is in good agreement with the previous studies[55]. With the electronic structure established, we applied finite difference method to calculate the $A_1$ and $B_2$ vibrational frequencies and modes which corresponds to the predominant peaks in the low-temperature Raman measurements.

To perform the Raman assignment and make meaningful comparisons between DFT and experiments, we carefully compare the $B_2$ vibrational modes of both AFM-*b* and AFM-*a* phases by ordering them in the ascending order of their vibrational frequencies. As shown in Supplementary Fig. 23, the vibrational eigen modes of 1 Q are in good correspondence between the AFM-*b* and AFM-*a* phases.

## Data availability

Supplementary Information is available for this paper. The Raman data generated in this study have been deposited in the Figshare database at https://doi.org/10.6084/m9.figshare.24022497. Additional data are available from the corresponding author upon request.

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

## Acknowledgements

H.W., Y.X., Y.W., Z.W., L-Q. C, I.D. and V.G. acknowledge the primary support from US department of Energy, Office of Science, Basic Energy Sciences, under Award Number DE-SC0020145 as part of the Computational Materials Science Program. H.P. and V.G. acknowledge support from the Department of Energy grant DE-SC0012375 for contributions to Raman analysis. R.Z., L.M., Z.M., and V.G. acknowledge support from the NSF MRSEC Center for Nanoscale Science grant number DMR-2011839 for partial contributions to crystal growth and performing the Raman experiments in the Materials Characterization Laboratory. Major support for crystal growth and characterization was provided by the National Science Foundation through the Penn State 2D Crystal Consortium-Materials Innovation Platform (2DCC-MIP) under NSF cooperative agreement DMR-1539916 and 2039351. G.H., G.B. and R.C. acknowledge support from the U.S. Department of Energy, Office of Science, Office of Basic Energy Sciences, Materials Sciences and Engineering Division, under Contract No. DE-AC02-05CH11231: Materials Project program KC23MP as well as from the Communauté Française de Belgique, grant ARC 18/23-093 for the phonon and electron-phonon theoretical work. First-principles calculations were carried out partially on the resources of NERSC supported by the Office of Science of the US Department of Energy under contract No. DE-AC02-05CH11231 (Y.W.).

## Author contributions

H.W. and V.G. conceived the project. Raman spectroscopy measurements and analysis were carried out by H.W., H.P., Z.R., M.W. and V.G. Density functional calculations were done by Y.X., Yi.W., Z.W., R.C., G.B., G.H., I.D., and L-Q. C. Crystal Growth and characterization were done by L.M., Yu.W. and Z.Q.M. The paper was written by H.W. and V.G. with inputs from all authors.

## Competing interests

The authors declare no competing interests.
