## [Peer Review File · Nature Communications]

REVIEWER COMMENTS

Reviewer #1 (Remarks to the Author):

The authors combined temperature-dependent Raman spectroscopy with first-principles calculations to reveal a strong coupling between two particular phonon modes and electronic/magnetic structures of $\text{Ca}_3\text{Ru}_2\text{O}_7$. They also provided experimental evidence for a phonon-assisted density wave fluctuation around the metal-pseudogap phase transition in $\text{Ca}_3\text{Ru}_2\text{O}_7$.

The results are interesting and the analysis is comprehensive. The work deserves publication in some form. However, I have some comments for the authors to address, which hopefully can further improve the quality of this manuscript.

1. The authors wrote that "The $B^A P_2$ modulation further compresses the RuO_6 octahedron along $d_{\text{perpendicular}}$ direction, while the $B^A M_2$ mode elongates it irrespective of the sign of the eigenvectors". From Fig. 3, it seems that the $B^A P_2$ phonon only involves the movement of oxygen atoms in the ab -plane, while the $B^A M_2$ phonon has oxygen movement out of the ab -plane. For the $B^A P_2$ phonon, I can understand why the d ratio (panel d) does not depend on the sign of phonon displacement. However, for the $B^A M_2$ phonon, if the oxygen atoms vibrate out of the ab -plane, how come the d ratio does not depend on the sign of phonon displacement? Could the authors clarify this point and also could the authors provide the numerical phonon eigenvectors of these two important phonons, in addition to the schematics shown in Fig. 3? Another related question is how the authors define the phonon displacement " u ". What is " $\sqrt{\text{amu}}$ " in the caption of Fig. 3d? Some explanation is needed.

2. The authors used the phrase "density wave fluctuations" instead of "a static density wave" because "it is largely incoherent and short-lived". Can the authors estimate the characteristic lifetime for the density wave fluctuations? How does it correlate with the phonon frequency of $B^A P_2$?

3. The authors used the word "pseudogap" phase throughout the paper. I am wondering whether it has exactly the same meaning as the one for cuprate superconductors. From Fig. 4a, it seems that the gap is momentum-dependent. However, in Fig. 4c and 4d, the pseudogap is extracted from the Raman response function which does not have momentum dependence. Could the authors clarify this point? How did the authors determine the pseudogap size and does the pseudogap have strong momentum dependence?

4. The authors corrected the phonon assignment in the previous study. In another previous study (Phys. Rev. Research 2, 023141 (2020)), the authors of that paper used the first-principles calculations and found that the low-temperature structure of Ca₃Ru₂O₇ has Pn2₁a space group, instead of Bb2₁m. More importantly, the theoretically predicted charge and spin density wave of Ca₃Ru₂O₇ is associated with that orthorhombic Pn2₁a structure. Could the authors comment on whether they found this Pn2₁a structure from their low-temperature Raman measurements?

Below are some more technical points:

5. The final self-consistent calculations and phonon calculations take into account the spin-orbit coupling (otherwise AFM-a and AFM-b can not be distinguished). I am wondering during the atomic position optimization, whether the authors also included the spin-orbit coupling. If yes, how different are the optimal crystal structures for AFM-a and AFM-b? And which state has lower energy?

6. This might be an issue of nomenclature. The authors used the phrase "electron-phonon coupling" or EPC. However, the unit they use is eV/Å. In my understanding, eV/Å is the unit for deformation potential "D". Electron-phonon coupling is usually dimensionless, and is proportional to the square of deformation potential (e.g. see Eq. (3) of PRL 86, 4366 (2001)). I suggest that the authors add some explanation on this point.

7. Many references miss the page number and it should be fixed.

Reviewer #2 (Remarks to the Author):

the authors did a prettyful work to investigate the electron phonon coupling in Ca₃Ru₂O₇, by combining the temperature dependent Raman scattering and SO coupling included DFT calculation. the Raman peaks are reattributed according to a new space group of the low temperature phase, and the strong electron phonon coupling with B2 phonon is identified both through the fitting of the Raman peaks and the DFT calculated ones. also the charge / spin density wave fluctuation has been proposed. the results are new and corrected the old Raman result interpretation and consitent with other reported results such as ARPES and transport. this work can have important compact on the research of the correlated matters through Raman light scattering. The manuscript is also well organized and written. overall, i suggest its acceptance by nature communications.

Reviewer #3 (Remarks to the Author):

The paper by Huaiyu (Hugo) Wang et al. presents a spontaneous Raman scattering study of phonon modes and electronic excitations in $\text{Ca}_3\text{Ru}_2\text{O}_7$. The authors argue that specific B2 phonon modes are strongly coupled to the material's pseudogap and mediate the formation of a charge/spin density wave order. These conclusions are supported by density functional theory calculations in the frozen phonon approximation.

The authors have vast expertise in spontaneous Raman scattering and optical spectroscopy. The data are of good quality and the analysis is systematic. However, the question of whether this paper is fit for publication in Nature Communications depends on the robustness and novelty of the results, as well as their broad impact on the condensed matter community. Unfortunately, I do not think that the present results meet the bar to satisfy these criteria. The paper would be more suitable in a journal for a specialized audience. Below, I describe my major concerns.

1) One of the paper's main results is that two phonon modes with B2 symmetry strongly couple to the pseudogap by opening or closing it. This conclusion was made through a complex fit of the Raman spectrum (Fig. 2a), in which the structure around 380-450 cm^{-1} is decomposed in a series of five phonon modes. It is hard to say that these results provide direct evidence for the authors' claim. First, the Raman feature of interest is very congested, and the possible individual phonons are hardly distinguishable in the absence of a mechanism that can disentangle them (e.g., the application of an external stimulus that tunes their frequencies or lowers the crystal symmetry). Therefore, the fit is very questionable. Second, using LDA+U+SOI calculations to assign the modes can also lead to errors: for example, the calculations are performed at zero temperature, and they are not robust across different functionals (e.g., PBE). How can the authors be sure that the LDA+U+SOI results truly describe their Raman spectra? The authors should cross-check their computational results with the experimental findings available in the literature. For example, a careful comparison of the electronic structure and phonon dispersion away from the Brillouin zone center could test the validity of the current computational approach.

2) The most impactful result of the paper is the putative discovery of short-lived charge/spin density wave fluctuations in this compound. Unfortunately, the claim is solely supported by a hump structure detected in the B2 electronic Raman symmetry channel (and absent in the A1 channel). As such, this is not the type of unambiguous evidence of density wave order that would grant publication in a high-impact factor journal such as Nature Communications.

3) Finally, the current version of the paper is tough to read: much useful information is hidden in the Supplementary Materials and continuous reference to these sections is made in the main text. The audience must read the Supplementary Information to understand the assignments and the data analysis made by the authors. As the paper is not written in a letter style, I strongly recommend that the authors reorganize their materials and guide the readers through their full spectra and analysis before focusing on the features of interest.

Response to Reviewer Comments

NCOMMS-22-53206-T: Strong Electron-Phonon Coupling driven Pseudogap Modulation and Density-Wave Fluctuations in a Correlated Polar Metal

Contents

- i. Response summary
- ii. Point-by-point response – Reviewer #1
- iii. Response to Reviewer #2
- iv. Point-by-point response – Reviewer #3

Response summary

We sincerely thank the reviewers for their thorough reading of our manuscript and for providing constructive feedback. Per their suggestions, we have carried out additional targeted experiments, computation, and analysis. These additions are:

(1) To disentangle the Raman peaks $B_2^{(12)}$ to $B_2^{(15)}$ via chemical pressure and provide further evidence for the peak assignments:

- Finding peaks in the 1800 grating $\text{Ca}_3\text{Ru}_2\text{O}_7$ B_2 Raman spectra by processing the raw Raman data.
- Temperature dependent Raman spectroscopy on 3% Ti doped $\text{Ca}_3(\text{Ru}_{1-x}\text{Ti}_x)_2\text{O}_7$.
- Different fitting models to demonstrate the robustness of fitting of the Raman peaks.
- Added new **Supplementary Note 5** and **Supplementary Figures 5-8** to summarize the new analysis that supports the assignment of five phonon modes in the B_2 spectra between 350 and 450 cm^{-1}
- Added a new **Figure 1** and updated main text to guide the readers through our full Raman spectra and analysis.

(2) To test the robustness of the DFT functional choice on the calculated energy of phonons.

- Sensitivity tests of phonon perturbed potential energy surface as a function of exchange-correlation functionals.
- Added a new **Supplementary Note 6** and **Supplementary Figures 9** to summarize the sensitivity tests of phonon energy calculations performed with different functionals (PBEsol, LDA, and PBE)

(3) To expand on the B_2^P and B_2^M eigen modes:

- Added a detailed description of the phonon eigen mode of B_2^P and B_2^M in the **Supplementary Note 9** with new **Supplementary Figures 15-16**.
- Added further analysis on the structure distortion induced by B_2^P and B_2^M perturbation in the **Supplementary Note 9** with new **Supplementary Figures 17-18**.

(4) To emphasize why the static density wave phase in $\text{Ca}_3\text{Ru}_2\text{O}_7$ is contradictory to existing X-ray data and demonstrate how B_2 phonon can perturb the structure to promote charge density wave fluctuation:

- Added a summary of recent high energy X-ray diffraction study on $\text{Ca}_3\text{Ru}_2\text{O}_7$ and B_2 phonon perturbed octahedra volume change in **Supplementary Note 14** and **Supplementary Figure 21-22**.

(5) To estimate the lifetime of observed phonon mediated charge density wave fluctuations:

- Added the systematics of fitting of the hump feature in B_2 electronic Raman spectra in **Supplementary Note 15** and **Supplementary Figure 23**.

Reviewer #1 (Remarks to the Author, Author Responses, and Actions Taken):

Reviewer Comment: The authors combined temperature-dependent Raman spectroscopy with first-principles calculations to reveal a strong coupling between two particular phonon modes and electronic/magnetic structures of Ca₃Ru₂O₇. They also provided experimental evidence for a phonon-assisted density wave fluctuation around the metal-pseudogap phase transition in Ca₃Ru₂O₇.

The results are interesting and the analysis is comprehensive. The work deserves publication in some form. However, I have some comments for the authors to address, which hopefully can further improve the quality of this manuscript.

Author Response: We are happy to hear that the reviewer finds our experimental and computational results interesting and the analysis comprehensive. We thank them for their careful reading of our manuscript.

Reviewer Comment: 1. The authors wrote that "The B₂^P modulation further compresses the RuO₆ octahedron along d_perpendicular direction, while the B₂^M mode elongates it irrespective of the sign of the eigenvectors". From Fig. 3, it seems that the B₂^P phonon only involves the movement of oxygen atoms in the ab-plane, while the B₂^M phonon has oxygen movement out of the ab-plane. For the B₂^P phonon, I can understand why the d ratio (panel d) does not depend on the sign of phonon displacement. However, for the B₂^M phonon, if the oxygen atoms vibrate out of the ab-plane, how come the d ratio does not depend on the sign of phonon displacement? Could the authors clarify this point and also could the authors provide the numerical phonon eigenvectors of these two important phonons, in addition to the schematics shown in Fig. 3? Another related question is how the authors define the phonon displacement "u". What is "sqrt(amu) in the caption of Fig. 3d"? Some explanation is needed.

Author Response:

Eigen Vectors: The eigen vectors for an amplitude of $1 \text{ \AA} \sqrt{\text{amu}}$ of B₂^P and B₂^M phonons are now listed in **Supplementary Figs. 15** and **16** with atomic positions of relevant Ru and O atoms and their atomic displacement vector corresponding to 1Q phonon displacement in units of Å. For a projection view of the two phonons in *a-b*, *a-c* and *b-c* planes, please refer to **Supplementary Fig. 25**.

B₂^P phonon eigen mode

Octahedral	Atom labels	Atomic position (Å)	Displacement vector (Å)
Oct_1	Ru	[1.369, 4.232, 7.873]	[-5.32e-3, 9.6e-4, 1.93e-3]
Oct_1	O(1)	[0.961, 4.104, 5.916]	[2.75e-2, -2.64e-2, -1.08e-2]
Oct_1	O(2)	[1.853, 4.362, 9.808]	[2.42e-2, 3.16e-2, 0]
Oct_1	O(3)_1	[-0.291, 5.353, 8.239]	[6.03e-2, 6.08e-2, -2.29e-2]
Oct_1	O(3)_2	[0.291, 2.533, 8.239]	[6.03e-2, -6.08e-2, 2.29e-2]
Oct_1	O(4)_1	[2.438, 5.901, 7.498]	[-5.30e-2, 5.08e-2, 3.87e-3]
Oct_1, Oct_2	O(4)_2	[2.990, 3.082, 7.498]	[-5.30e-2, -5.08e-2, -3.87e-3]
Oct_2	Ru	[4.059, 1.413, 7.873]	[-5.32e-3, -9.6e-4, -1.93e-3]
Oct_2	O(1)	[4.467, 1.285, 5.916]	[2.75e-2, 2.64e-2, 1.08e-2]
Oct_2	O(2)	[3.575, 1.543, 9.808]	[2.42e-2, -3.16e-2, 0]
Oct_2	O(3)_1	[5.137, -0.286, 8.239]	[6.03e-2, 6.08e-2, -2.29e-2]
Oct_2	O(3)_2	[5.720, 2.533, 8.239]	[6.03e-2, -6.08e-2, 2.29e-2]
Oct_2	O(4)_1	[2.438, 0.263, 7.498]	[-5.30e-2, 5.08e-2, 3.87e-3]

Supplementary Figure 15. Eigen vector of B₂^P phonon. The 1 Q (Å√amu) eigen vector of B₂^P phonon with initial atomic position of moving atoms respectively.

B₂^M phonon Eigen mode

Octahedral	Atom labels	Atomic position (Å)	Displacement vector (Å)
Oct_1	Ru	[1.369, 4.232, 7.873]	[3.44e-4, 1.25e-2, 1.19e-3]
Oct_1	O(1)	[0.961, 4.104, 5.916]	[2.24e-2, -5.38e-2, -1.41e-2]
Oct_1	O(2)	[1.853, 4.362, 9.808]	[7.62e-2, -9.23e-2, 0]
Oct_1	O(3)_1	[-0.291, 5.353, 8.239]	[-2.73e-2, -2.25e-2, 2.55e-2]
Oct_1	O(3)_2	[0.291, 2.533, 8.239]	[-2.73e-2, 2.25e-2, -2.55e-2]
Oct_1	O(4)_1	[2.438, 5.901, 7.498]	[-2.49e-2, 8.33e-3, -3.27e-2]
Oct_1, Oct_2	O(4)_2	[2.990, 3.082, 7.498]	[-2.49e-2, -8.33e-3, 3.27e-2]
Oct_2	Ru	[4.059, 1.413, 7.873]	[3.44e-4, -1.25e-2, -1.19e-3]
Oct_2	O(1)	[4.467, 1.285, 5.916]	[2.24e-2, 5.38e-2, 1.41e-2]
Oct_2	O(2)	[3.575, 1.543, 9.808]	[7.62e-2, 9.23e-2, 0]
Oct_2	O(3)_1	[5.137, -0.286, 8.239]	[-2.73e-2, -2.25e-2, 2.55e-2]
Oct_2	O(3)_2	[5.720, 2.533, 8.239]	[-2.73e-2, 2.24e-2, -2.55e-2]
Oct_2	O(4)_1	[2.438, 0.263, 7.498]	[-2.49e-2, 8.33e-2, -3.27e-2]

Supplementary Figure 16. Eigen vector of B₂^M phonon. The 1 Q (Å√amu) eigen vector of B₂^M phonon with initial atomic position of moving atoms respectively.

Definition of u : We represent phonons using mass-weighted atomic displacements $Q_{I\alpha} \equiv \sqrt{m_I}u_{I\alpha}$, which are the canonical coordinates to express the dynamical matrix \mathbf{D} :

$$D_{I\alpha,J\beta} = \frac{1}{\sqrt{m_I m_J}} \frac{\partial^2 E}{\partial u_{I\alpha} \partial u_{J\beta}} = \frac{\partial^2 E}{\partial Q_{I\alpha} \partial Q_{J\beta}},$$

where I is the atom index, α is the space direction ($\alpha = x, y,$ and z), m_I is the mass of atom I (in atomic units of mass, a.m.u = $1.6605 \cdot 10^{-24}$ g), and $u_{I\alpha}$ is the displacement of atom I along direction α (in Å). By diagonalizing the dynamical matrix, one obtains the resonant frequencies ω_n (in inverse atomic units of time) and the normal modes \mathbf{e}_n (which are adimensional vectors):

$$\sum_{j\beta} D_{I\alpha,J\beta} e_{j\beta,n} = \omega_n^2 e_{I\alpha,n}.$$

Therefore, the atomic displacements can be decomposed along the normal modes as

$$u_{I\alpha} = \frac{Q_{I\alpha}}{\sqrt{m_I}} = \frac{1}{\sqrt{m_I}} \sum_n Q_n e_{I\alpha,n},$$

where $Q_n \equiv \sum_{I\alpha} Q_{I\alpha}^* e_{I\alpha,n}$ denotes the coordinate of the n^{th} normal mode [in (a.m.u)^{1/2} Å]. We updated the method section of the manuscript and the figures to clarify the definitions of $u_{I\alpha}$, $Q_{I\alpha}$, and Q_n , ensuring that notations are internally consistent and in conformity with literature conventions.

The dependence of the d_{\perp}/d_{\parallel} and the in-plane bond angle (θ) on the sign of the phonon displacement: Thank you for this query. We have added a revised **Supplementary Note 9** dedicated to this discussion. The unit cell $\text{Ca}_3\text{Ru}_2\text{O}_7$ hosts two sets of RuO_6 octahedra that change in different ways when perturbed by the B_2 phonons: Octahedra 1 (in blue) and Octahedra 2 (in red). **Figures 4d and 4e** are plots of d ratio and in-plane bond angles averaged between two sets of octahedra. The B_2^{P} and B_2^{M} phonons would asymmetrically change the d_{\perp}/d_{\parallel} and θ of individual octahedra, while the average changes are symmetric (**Supplementary Fig. 17**). A more detailed discussion of these figures is given now in **Supplementary Note 9**.

Supplementary Figure 17. d ratio and in-plane bond angles of octahedra 1 and 2 modulated by B_2^{M} and B_2^{P} phonons. The ratio of RuO_6 octahedra cage apical Ru-O bond length (d_{\perp}) over in-plane bond length (d_{\parallel}) calculated at two neighboring octahedra modulated by (a) B_2^{M} and (b) B_2^{P} phonon. Ru-O-Ru in-plane bond angles modulated by (c) B_2^{M} and (d) B_2^{P} phonon. The labeling refer to illustration in **Supplementary Fig. 15** and **16**.

The B_2^{P} phonon promotes in-plane charge hopping between neighboring RuO_6 octahedra sites by coherently modulating hopping parameters (**Supplementary Fig. 18a**). On the other hand, B_2^{M}

phonon cannot modulate hopping parameters due to competing effects from bond length and bond angle change (**Supplementary Fig. 18b**).

Supplementary Figure 18. In-plane hopping integral modulated by B_2^P and B_2^M phonon. (a) The real-space Ru-O bond length change (color code: red means increase, and blue means decrease) and in plane Ru-O-Ru bond angle change ($\theta \uparrow$ means angle increase, and $\theta \downarrow$ means angle decrease) after 1Q B_2^P phonon modulation. This phonon increases the hopping integral t either along the $[1\ 1\ 0]$ or along the $[1\ \bar{1}\ 0]$ crystallographic direction. (b) The real-space Ru-O bond length change (color code: red means increase, and blue means decrease) and in plane Ru-O-Ru bond angle change ($\theta \uparrow$ means angle increase, and $\theta \downarrow$ means angle decrease) after 1Q B_2^M phonon modulation. The change in the hopping integral t along any of the directions is not monotonic due to competing effects from bond length changes and θ changes after 1Q B_2^M phonon displacement.

Action Taken: The above discussions and clarifications, along with the mentioned figures are now included in the manuscript.

Reviewer Comment: 2. The authors used the phrase "density wave fluctuations" instead of "a static density wave" because "it is largely incoherent and short-lived". Can the authors estimate the characteristic lifetime for the density wave fluctuations? How does it correlate with the phonon frequency of B_2^P ?

Author Response: We thank the reviewer for raising this important question.

Static versus Dynamic Density Wave: A static density wave in $\text{Ca}_3\text{Ru}_2\text{O}_7$ is predicted to modify the space group of the crystal¹. However, with the most recent high energy X-ray pair distribution function study on $\text{Ca}_3\text{Ru}_2\text{O}_7$ ², conducted with a larger precision on Ru-O bond length than required to resolve the proposed static density wave phase, the space group remains $\text{Bb}2_1\text{m}$, and this indicates the static density wave distortion is not observed in the low temperature. We have added a **Supplementary Note 14** titled "lack of static charge density wave phase in $\text{Ca}_3\text{Ru}_2\text{O}_7$ from structural evidence", to elaborate on these studies.

Characteristic Lifetime of the Density Wave Fluctuations: The FWHM of the hump feature in **Fig. 5c** can have multiple origins, such as inhomogeneous broadening from defects or various sizes of density wave domains. This makes accessing the intrinsic lifetime of the density wave fluctuation challenging. As a very rough estimate, the rising edge of the hump (see

Supplementary Fig. 23a, reproduced below) can be well described by a Gaussian fitting which is estimated to be 135cm^{-1} at 18K, which translates to around 250fs in coherence lifetime. The FWHM of density wave fluctuation feature increases (and hence the lifetime decreases) as the temperature increases (**Supplementary Fig. 23b**), which has the same trend as the energy of the B_2^P phonon (**Supplementary Fig. 23c**). The frequency of the B_2^P phonon has a temporal period of $\sim 83\text{fs}$ at 18K, which is a third of the lifetime of the density wave fluctuation. This suggests that these phonons can indeed mediate the density wave fluctuations on similar timescales. To further study the lifetime of density wave fluctuations, we suggest future experiments such as ultrafast pump probe to accurately characterize both the temporal and spatial coherence of the density wave fluctuation.

Action Taken: The above discussion has been added to **Supplementary Note 14** and **Supplementary Note 15**.

Supplementary Fig. 23 Temperature-dependence of density wave fluctuation lifetime compared with phonon energy of B_2^P phonon. (a) The Gaussian fitting profile (shaded area) overlaid on top of the subtracted electronic Raman response $\Delta\chi''_{B_2}(\omega, T)$ after subtracting the response at 49K (dotted line). The temperature of the collected electronic Raman response is labeled in each graph. (b) The temperature dependence of the FWHM of the fitted Gaussian profile. (c) The phonon energy of the B_2^P phonon as a function of the temperature.

Reviewer Comment: 3. The authors used the word "pseudogap" phase throughout the paper. I am wondering whether it has exactly the same meaning as the one for cuprate superconductors. From Fig. 4a, it seems that the gap is momentum - dependent. However, in Fig. 4c and 4d, the pseudogap is extracted from the Raman response function which does not have momentum dependence.

Could the authors clarify this point? How did the authors determine the pseudogap size and does the pseudogap have strong momentum dependence?

Author Response:

Definition of pseudogap and comparison to superconductors: The pseudogap is a partial gap that opens in their spectral function at specific k points. The definition of the pseudogap is the same for $\text{Ca}_3\text{Ru}_2\text{O}_7$ and for the superconducting cuprates. However, the pseudogap phase in cuprates implies a precursor to superconductivity due to preformed cooper pairs³. The pseudogap phase in $\text{Ca}_3\text{Ru}_2\text{O}_7$ has not been linked to superconductivity. While the formation of pseudogap phase in $\text{Ca}_3\text{Ru}_2\text{O}_7$ is not well understood, the pseudogap phase in $\text{Ca}_3\text{Ru}_2\text{O}_7$ is unlikely of the same physical origin as that in cuprates.

Momentum dependence of the pseudogap: The pseudogap in $\text{Ca}_3\text{Ru}_2\text{O}_7$ is weakly momentum dependent below the pseudogap transition temperature of 48K with the smallest gap (around 8meV between highest occupied band to the fermi level opened at $(\pm \frac{1}{2}\pi, \pm \frac{1}{2}\pi)$) and the largest gap around $\Gamma(0,0)$ at ~10meV between highest occupied band to the fermi level, and this can be observed in one previous ARPES study on $\text{Ca}_3\text{Ru}_2\text{O}_7$ ⁴. It should be emphasized that vast majority of Fermi-surface is gapped below $T_c=48\text{K}$ and the gapped surface (except the smallest gap at $(\pm \frac{1}{2}\pi, \pm \frac{1}{2}\pi)$) is not strongly momentum dependent since the gap size remains largely constant across grey area marked in **Fig. 5a** inset (see for example Fig. 3 in this ARPES work⁴.)

How the pseudogap was determined: **Figure 5, panels c and d** show how we estimated the pseudogap in this work. The Raman background after subtracting the Raman peaks and the reference background response at 49K, we obtain the spectra shown in **Figs. 5c, d** for the B_2 and A_1 measurement geometries, respectively. In Raman spectroscopy, we are observing an averaged pseudogap size weighted by the Raman Vertex as indicated by the red/blue colormaps in the **Fig. 4a** inset. After careful review, we realized the marking of pseudogap and CDW fluctuation energy scale is wrong and corrected the label in the new **Fig. 4c and 4d**. The plot of Raman response function $\chi''_u(\omega)$ should not be confused with $\sigma(\omega)$ from FTIR measurement. The former is a weighted two-particle correlation function $\chi''_u(\omega) = \langle \rho_u(\omega)\rho_u(-\omega) \rangle$, which is the correlation between electron and hole in the excited state. The latter is single particle excitation spectra. Thus, the correct reading of the pseudogap and CDW energy scale is finding the peak value of the spectral weight change. By fitting a Gaussian peak to the dip features below 400 cm^{-1} arising from the pseudogap opening, we estimate the pseudogap size as shown.

Action taken: We have briefly captured the above discussion on the pseudogap in the main text. In addition, we have made new **Fig. 5c and 5d**. We added the following sentences to the main text to reflect the above discussion: “From Gaussian fitting of the pseudogap dip feature, the probed pseudogap size is $2\Delta_{\text{pseudogap}} = 22.5 \pm 1.4 \text{ meV}$ from B_2 Raman response function and $2\Delta_{\text{pseudogap}} = 21.4 \pm 1.3 \text{ meV}$ from the A_1 Raman function at 28K. This agrees well with the averaged pseudogap size extracted from the optical conductivity measurement⁵, which was reported to be ~20meV.”

Reviewer Comment: 4. The authors corrected the phonon assignment in the previous study. In another previous study (Phys. Rev. Research 2, 023141 (2020)), the authors of that paper used the

first-principles calculations and found that the low-temperature structure of $\text{Ca}_3\text{Ru}_2\text{O}_7$ has $\text{Pn}2_1\text{a}$ space group, instead of $\text{Bb}2_1\text{m}$. More importantly, the theoretically predicted charge and spin density wave of $\text{Ca}_3\text{Ru}_2\text{O}_7$ is associated with that orthorhombic $\text{Pn}2_1\text{a}$ structure. Could the authors comment on whether they found this $\text{Pn}2_1\text{a}$ structure from their low-temperature Raman measurements?

Author Response: Raman process is only sensitive to the point group of the crystal. Since both space groups belong to a $m2m$ point group, the Raman spectroscopy cannot unfortunately distinguish between the two-space groups. Diffraction experiments are therefore necessary.

The DFT study (Phys. Rev. Research 2, 023141 (2020)) suggests that a key experiment is to locate (011) Bragg peak from diffraction measurement at low temperature. However, a closer look at the reported cif file in that work reveals that the intensity of (011) Bragg peak is ~ 40000 weaker than that of (111) or (004), and thus requires a very high signal to noise ratio of 92 dB, which is unlikely to be achieved in diffraction experiments.

On the other hand, techniques such as PDF with high X-ray diffraction can give rise to accurate bond length measurement. The recent X-ray PDF study² of $\text{Ca}_3\text{Ru}_2\text{O}_7$ reports the Ru-O bond length with an error bar of 0.01 Å. The reported structure distortion that induces the static charge density wave phase by the DFT study¹ result in a bond length difference of the neighboring in-plane Ru-O bond as large as 0.07 Å, which one should have been able to be resolve in the X-ray study. Nonetheless, the experimental PDF study reported that $\text{Bb}2_1\text{m}$ space group shows a reasonable fit down to 10K, thus most likely ruling out the $\text{Pn}2_1\text{a}$ phase.

The DFT study predicting $\text{Pn}2_1\text{a}$ is nonetheless potentially insightful in that this phase exhibits a bifurcation of the neighboring octahedral volume. Excitation of B_2 phonons can modulate the neighboring octahedra volumes in a similar manner (see **Supplementary Fig. 22** below, using B_2^{P} as an example, but the phenomenon is universal to all B_2 modes). This makes the excitation of B_2 phonons as a potential pathway to stabilize a dynamic density wave phase in $\text{Ca}_3\text{Ru}_2\text{O}_7$.

Supplementary Fig. 22 Volume perturbation by B_2^P phonon mode. The volume of Octahedra 1 and Octahedra 2 as a function of B_2^P phonon displacement.

Action Taken: A brief note on the absence of the predicted Pn21a is now included in the main text as well as in **Supplementary Note 14**.

Reviewer Comments: Below are some more technical points:

5. The final self-consistent calculations and phonon calculations take into account the spin-orbit coupling (otherwise AFM-a and AFM-b can not be distinguished). I am wondering during the atomic position optimization, whether the authors also included the spin-orbit coupling. If yes, how different are the optimal crystal structures for AFM-a and AFM-b? And which state has lower energy?

Author Response: We thank the reviewer for pointing out this omission. The spin-orbit coupling was indeed included for the atomic position optimizations while the lattice parameters were fixed to experimental measurements, as detailed in the main text. Our results show that AFM-b is more stable than AFM-a (with an energy difference of 17.3 meV).

Action taken: We have modified the **Methods** section to clarify this aspect: “The spin-orbit interaction is included for the atomic position optimizations. The lattice parameters in all the DFT studies are from the reported experimental structures with a space group of $Bb21m^6$. Our results show that AFM-b is more stable than AFM-a (with an energy difference of 17.3 meV).”

Reviewer Comment: 6. This might be an issue of nomenclature. The authors used the phrase "electron-phonon coupling" or EPC. However, the unit they use is eV/A. In my understanding,

eV/A is the unit for deformation potential "D". Electron-phonon coupling is usually dimensionless, and is proportional to the square of deformation potential (e.g. see Eq. (3) of PRL 86, 4366 (2001)). I suggest that the authors add some explanation on this point.

Author Response: We thank the reviewer for pointing this out. We have updated the main text to clarify our definition on the "EPC strength" with the above-mentioned reference. In the manuscript, the D is indeed the deformation potential instead of the unitless mass enhancement parameter λ . Some papers^{7,8} also refer to the deformation potential as an indication of the EPC strength.

Action taken: We have changed references to the "EPC strength" to "deformation potential" throughout the document to address this point.

Reviewer Comments: 7. Many references miss the page number and it should be fixed.

Author Response and Action taken: Thank you for point this out. We have fixed the references and added the missing page numbers.

Reviewer #2 (Remarks to the Author, Author Responses, and Actions Taken):

Reviewer Comment: The authors did a prettyful work to investigate the electron phonon coupling in $\text{Ca}_3\text{Ru}_2\text{O}_7$, by combining the temperature dependent Raman scattering and SO coupling included DFT calculation. the Raman peaks are reattributed according to a new space group of the low temperature phase, and the strong electron phonon coupling with B_2 phonon is identified both through the fitting of the Raman peaks and the DFT calculated ones. also the charge / spin density wave fluctuation has been proposed. the results are new and corrected the old Raman result interpretation and consistent with other reported results such as ARPES and transport. this work can have important impact on the research of the correlated matters through Raman light scattering. The manuscript is also well organized and written. overall, i suggest its acceptance by nature communications.

Author Response: We are happy to hear that the reviewer finds our work impactful and we appreciate their positive comments. We do hope our work can contribute to a deeper understanding of the rich spectrum of physical phenomena in ruthenates.

Reviewer #3 (Remarks to the Author, Author Response, and Action taken):

Reviewer Comments: The paper by Huaiyu (Hugo) Wang et al. presents a spontaneous Raman scattering study of phonon modes and electronic excitations in $\text{Ca}_3\text{Ru}_2\text{O}_7$. The authors argue that specific B_2 phonon modes are strongly coupled to the material's pseudogap and mediate the formation of a charge/spin density wave order. These conclusions are supported by density functional theory calculations in the frozen phonon approximation.

The authors have vast expertise in spontaneous Raman scattering and optical spectroscopy. The data are of good quality and the analysis is systematic. However, the question of whether this paper is fit for publication in Nature Communications depends on the robustness and novelty of the results, as well as their broad impact on the condensed matter community. Unfortunately, I do not think that the present results meet the bar to satisfy these criteria. The paper would be more suitable in a journal for a specialized audience. Below, I describe my major concerns.

Author Response: We deeply appreciate the time taken by the reviewer to share their constructive comments which have significantly helped in improving the arguments and clarity of the manuscript. As seen below, we have performed additional experiments, analysis and DFT calculations to address these comments thoroughly. We do hope that our detailed responses below and the revised manuscript can convince the reviewer to reconsider their decision. We do believe that our work is of broad interest to researchers in the condensed matter and spectroscopy community and motivates further studies on 4d ruthenates that are far less explored than 3d or 5d compounds. This work straightens out decades of incorrect spectroscopy literature on this ruthenate that provides new insights and should lead to a more robust progress going forward.

Reviewer Comments: 1) One of the paper's main results is that two phonon modes with B_2 symmetry strongly couple to the pseudogap by opening or closing it. This conclusion was made through a complex fit of the Raman spectrum (Fig. 2a), in which the structure around $380\text{-}450\text{ cm}^{-1}$ is decomposed in a series of five phonon modes. It is hard to say that these results provide direct evidence for the authors' claim. First, the Raman feature of interest is very congested, and the possible individual phonons are hardly distinguishable in the absence of a mechanism that can disentangle them (e.g., the application of an external stimulus that tunes their frequencies or lowers the crystal symmetry). Therefore, the fit is very questionable.

Author Response: This is an excellent question which deserves a rigorous explanation in the manuscript. Below we show that (1) Even the raw Raman data reveals the presence of five peaks under the broad Raman peak in question. (2) Following reviewer's suggestion for an "external stimulus", by a slight addition of 3% Ti, these multiple components of the broad peak become far clearer and more distinct. (3) We demonstrate that our fitting model results in better fit of our data than that proposed by previous literature.

(1) Peaks identification from raw Raman data:

The analysis below is free of peak fitting and should provide further confidence in our claim of five peaks between 350 and 500 cm^{-1} . The first and second derivatives of a function can sensitively reveal the presence of a peak as illustrated by a simple Lorentzian in **Supplementary Figs 5, a-c**, reproduced below. The $\text{Ca}_3\text{Ru}_2\text{O}_7$ B_2 Raman profile at 10K (**Supplementary Fig. 5d**) has many

peaks between 350 and 500 cm^{-1} . The first derivative of the experimental Raman data (**Supplementary Fig. 5e**) reveals several positions. Many of the first derivatives do cross zero, indicating that there are peaks in the Raman spectra; however, three of them do not cross zero (~ 388 , 403 and 440 cm^{-1}). The second derivative of the Raman data (**Supplementary Fig. 5f**) further reveals dips at these three Raman shift positions, confirming that they are shoulder peaks and should be considered during fitting. The identified peaks agree well with our fitting functions choice. **Supplementary Fig. 5g** below shows that this method of analyzing the raw spectra with no fitting still shows a good agreement with the temperature dependence of the Raman peaks extracted from peak fitting. This analysis provides strong confidence in these results.

Supplementary Fig. 5 Finding peaks from processing Raman raw data. (a) Lorentzian peak profile. (b) First derivative of peak intensity with respect to peak position. (c) Second derivative of peak intensity with respect to peak position. (d) $\text{Ca}_3\text{Ru}_2\text{O}_7$ B_2 Raman profile at 10K. (e) The first derivative of data in (d) with respect to Raman shift. (f) The second derivative of data in (d) with respect to Raman shift. The black arrows indicate the feature of a peak. (g) The peak finding method has been applied to 1800 grating data (red) to obtain the energy of $B_2^{(11)}$ – $B_2^{(15)}$ phonon peaks. The black data points are phonon energy obtained from peak fitting of 600 grating data points.

(2) Peaks identification from Raman data on 3%Ti doped $\text{Ca}_3\text{Ru}_2\text{O}_7$:

We follow the reviewer’s suggestion to use external stimulus to disentangle the peaks by tuning the chemical potential in $\text{Ca}_3\text{Ru}_2\text{O}_7$ via doping 3% Ti to replace some of the Ru sites. The doping of Ti does not involve a symmetry change, with the space group remaining $Bb2_1m$ after doping⁹. Since there is no crystallographic point group change before and after Ti doping, the number of phonon modes in B_2 irrep should remain the same between pure vs. the Ti-doped crystal structures. However, the magnetic structure at 10K change from AFM-b phase in pure $\text{Ca}_3\text{Ru}_2\text{O}_7$ to G-AFM phase in 3% Ti doped $\text{Ca}_3(\text{Ru}_{1-x}\text{Ti}_x)_2\text{O}_7$ (**Supplementary Fig. 6a**).

The congested profile between 350 to 500 cm^{-1} in AFM-b phase (**Supplementary Fig. 6b**) is now split into four distinct peaks in the G-AFM phase (**Supplementary Fig. 6c**), providing evidence that further supports our identification of several phonon peaks in this region (A fifth peak, if present, appears to have very low intensity in this doped sample). Furthermore, we examine the fitting across the measured Raman shift range up to 800 cm^{-1} and we can find one-to-one correspondence between the B_2 modes in pure $\text{Ca}_3\text{Ru}_2\text{O}_7$ phases, and those in 3% Ti doped $\text{Ca}_3\text{Ru}_2\text{O}_7$ phases (**Supplementary Fig. 7**). The doping effect on phonon energy is quite interesting and we plan to further study it a future study.

Supplementary Fig. 6 Tuning the peak frequency by doping Ti atoms. (a) Phase diagram of temperature and Ti doping percentage in $\text{Ca}_3\text{Ru}_2\text{O}_7$, adapted from reference⁹. The fitting results of individual B_2 phonon peaks (black solid line) overlaid with overall fitting results (blue solid line), background subtraction (grey filled area, see **Supplementary Note 4**) and raw data (blue dots) of (b) pure $\text{Ca}_3\text{Ru}_2\text{O}_7$ at 10K, and (c) 3% Ti doped $\text{Ca}_3\text{Ru}_2\text{O}_7$ at 10K.

Supplementary Fig. 7 Phonon assignment in pure and Ti doped $\text{Ca}_3\text{Ru}_2\text{O}_7$. The fitting results of individual B_2 phonon peaks (black solid line) overlaid with overall fitting results (blue solid line), background subtraction (grey filled area, see **Supplementary Note 4**) and raw data (blue dots) of (a) 3% Ti doped $\text{Ca}_3\text{Ru}_2\text{O}_7$ at 10K, (b) pure $\text{Ca}_3\text{Ru}_2\text{O}_7$ at 10K, (c) 3% Ti doped $\text{Ca}_3\text{Ru}_2\text{O}_7$ at 50K (AFM-a phase), (d) pure $\text{Ca}_3\text{Ru}_2\text{O}_7$ at 50K (AFM-a phase). The summary of B_2 phonon mode energy in pure and 3% Ti doped $\text{Ca}_3\text{Ru}_2\text{O}_7$.

(3) Robustness to different Raman background analysis:

The robustness of the five peak positions to different background subtraction schemes is discussed in **Supplementary Note 4**. If we assume minimal hybridization between phonons and electronic background, then the fitting result (see **Supplementary Fig. 8a**) is poor at near 360 cm^{-1} . We also must address the non-constant Raman background from electron-phonon coupling by considering a distorted profile (see **Supplementary Fig. 8b** and **Supplementary Note 4**). Since the affected area is composed of five peaks (see arguments above), using Fano-line shapes on five peaks leads to overfitting. We therefore make use of background subtraction to account for the electron-phonon coupling contribution in the Raman background. We also demonstrate the fitting result using a single Fano line shape to describe Raman profile between 380-450 cm^{-1} following the previous Raman paper¹⁰(see **Supplementary Fig. 8c**); this fitting quality is rather poor as highlighted by black arrows.

Supplementary Fig. 8 Fitting results of 10K $\text{Ca}_3\text{Ru}_2\text{O}_7$ B_2 Raman spectra with different models. The fitting results of individual phonon peaks (black solid line) overlaid with overall fitting results (blue solid line), background subtraction (grey filled area, see **Supplementary Note 4**) and raw data (blue dots) for (a) five phonon peaks between 350 and 500 cm^{-1} with small phonon peak electronic continuum mixture in background and (b) large mixture in background, and (c) single Fano peak between 350 and 500 cm^{-1} . The arrows indicate where the fitting is poor.

Action taken: We have included the above discussion in the updated **Supplementary Note 5**.

Reviewer comments: 1) Second, using LDA+U+SOI calculations to assign the modes can also lead to errors: for example, the calculations are performed at zero temperature, and they are not robust across different functionals (e.g., PBE). How can the authors be sure that the LDA+U+SOI results truly describe their Raman spectra? The authors should cross-check their computational results with the experimental findings available in the literature. For example, a careful comparison of the electronic structure and phonon dispersion away from the Brillouin zone center could test the validity of the current computational approach.

Author Response:

(1) Temperature dependence of the Raman modes: The reviewer is correct in noting that DFT calculations are performed at 0 K. As noted in the literature, first-principles calculations of phonon energies at 0 K have been used to compare with experimental data at finite temperatures^{11,12}. This comparison assumes that the effects of temperature on high-energy phonons are minimal, especially when the temperature change is within 50 K, as in our case. Previous work on perovskite oxides also shows that the high-energy optical branches demonstrate very limited temperature effects at elevated temperatures (as highlighted in Figure 1(d) in this reference¹³). In our subsequent sensitivity analysis of functionals (see **Supplementary Fig. 9**, reproduced below), the

potential energy surfaces are successfully fitted using second-order polynomial terms without including higher-order terms. Lack of anharmonic term in the phonon perturbed potential energy indicates a minimal temperature effect on the B_2 optical phonons of interest¹⁴. Consequently, we believe the optical phonon energies of AFM-*a* and AFM-*b* are mainly affected by the temperature-dependent magnetic phase transition, which is intrinsically accounted for in the DFT simulations with the spin-orbit coupling. From an experimental standpoint, the energy of phonons in the AFM-*b* phase at 13K can be extrapolated to lower temperatures. The observed phonon-phonon scattering is minimal, as indicated by the peak position of the Raman data collected above 48K, which does not change significantly up to 300K.

Supplementary Fig. 9 Sensitivity tests as a function of the exchange-correlation functionals (PBEsol, LDA, and PBE) and their impact on the potential energy surface ΔU with respect to the normal mode coordinate Q . (a) AFM-*a* $B_2^{(13)}$, (b) AFM-*a* $B_2^{(15)}$, (c) AFM-*b* $B_2^{(13)}$, and (d) AFM-*b* $B_2^{(15)}$.

(2) Testing Sensitivity to Functionals: We thank the reviewer for the comments on the sensitivity of the functionals. To address potential uncertainties, we have investigated two additional functionals: the PBE functional (as proposed by the reviewer) and the PBEsol functional (that is optimized for solid-state properties). We computed the potential energy surfaces (PES) by displacing the atoms along the $B_2^{(13)}$ and $B_2^{(15)}$ phonon modes and fitted the PES with quadratic functions. For consistency, on-site Hubbard repulsion of 1.2 eV on Ru 4*d* orbitals and spin-orbit coupling are included throughout. For ease of comparison, we reference the energy with respect to the ground-state energy of AFM-*b*. The results are summarized in **Supplementary Fig. 9**. For the phonon modes of interest, the PES is relatively insensitive to the choice of functionals as the coefficients of the quadratic terms show small variations across the functionals. More importantly, $B_2^{(15)}$ consistently shows larger coefficients compared to $B_2^{(13)}$, which indicates that the phonon frequency of $B_2^{(15)}$ would be larger than $B_2^{(13)}$ regardless of the choice of functionals. Therefore, it is expected that the relative order of $B_2^{(13)}$ and $B_2^{(15)}$ will remain unaltered and would not impact

our phonon mode assignments. We also highlighted that LDA has been shown to perform well for vibrational properties for oxides^{15,16}.

To the best of our knowledge, we are not aware of any phonon dispersion data published in $\text{Ca}_3\text{Ru}_2\text{O}_7$. For the electronic structure dispersion, our electronic band structure results in **Supplementary Fig. 25** are consistent with the DFT results in existing literature¹⁷, which has been compared to ARPES data¹.

Action Taken: We have added **Supplementary Note 6** and updated the main text to reflect the above discussion: “For two phonon modes of interest, $B_2^{(13)}$ and $B_2^{(15)}$, the potential energy surface change is insensitive to the choice of functionals (see **Supplementary Note 6**).”

Reviewer Comment: 2) The most impactful result of the paper is the putative discovery of short-lived charge/spin density wave fluctuations in this compound. Unfortunately, the claim is solely supported by a hump structure detected in the B2 electronic Raman symmetry channel (and absent in the A1 channel). As such, this is not the type of unambiguous evidence of density wave order that would grant publication in a high-impact factor journal such as Nature Communications.

Author Response: We are glad that the reviewer finds the result on density wave fluctuations impactful. Below we first present our case for why the Raman analysis in our work is also impactful, and then present our views on the density wave fluctuations.

(1) Impact of getting the phonon spectra right: Despite two decades of exploration, even the phonon assignment in literature in $\text{Ca}_3\text{Ru}_2\text{O}_7$ is questionable since even the crystallography was wrong in the previous assignments. Since electron-phonon coupling is a very central component of these strongly correlated systems, getting the phonons right is crucial to the physics that follows. Combined with spin-orbital coupling included DFT+U calculation, we reveal important role of electron-phonon coupling and provide a potential path for phononic control of electronic state in $\text{Ca}_3\text{Ru}_2\text{O}_7$. In this regard, we feel that this work is foundational for much research on ruthenates that will follow in these systems. We urge the reviewer to reconsider the impact of this work in this light.

(2) Density Wave Fluctuations: We appreciate the reviewer’s critique on the evidence of density wave order. Static charge density wave is excluded from X-ray experiment (See **Supplementary Note 14**). Density wave fluctuation without phonon assistance can be observed in diffraction study¹⁸, ARPES¹⁹ study in other compounds. Such evidence is systematically missing in $\text{Ca}_3\text{Ru}_2\text{O}_7$ with one recent ARPES²⁰ specifically commented “density-wave orders breaking translationa symmetry are, however, excluded since reconstruction preserves the original Brillouin-zone boundaries”. Electronic Raman has been used extensively to study phase competitions such as density wave fluctuation in correlated compounds such as cuprates²¹⁻²³, iron pnictide compounds²⁴, and so on.

We propose a phonon mediated density wave fluctuation, rather than a pre-existing density wave fluctuation. This is based on a robust symmetry signature in the background Raman. We rigorously use group theory to develop a potential mechanism that can possibly explain this symmetry selection rule. This is precisely what we have done here.

The proposed mechanism in **Fig. 5g** is optical selection rule forbidden unless assisted by a B₂ phonon. The results presented in **Fig. 5c** and **d** are consistent with the previous optical conductivity work⁵ and provides crucial new information on the physics. The mechanism presented in **Fig. 5g** is derived from the proposed mechanism in the previous optical conductivity work⁵ and we follow the mechanism to get the energy scale of the hump feature correct (see **Supplementary Note 11**). However, this mechanism of intersite Ru d electron hopping is optical selection rule forbidden (see **Supplementary Note 13**), and our data further supports the idea that vibronic coupling of B₂ phonons and d electrons allows the intersite Ru d electron hopping to take place. These new mechanistic insights are built on experimental spectroscopic observations that have robust symmetry signatures.

Reviewer Comment: 3) Finally, the current version of the paper is tough to read: much useful information is hidden in the Supplementary Materials and continuous reference to these sections is made in the main text. The audience must read the Supplementary Information to understand the assignments and the data analysis made by the authors. As the paper is not written in a letter style, I strongly recommend that the authors reorganize their materials and guide the readers through their full spectra and analysis before focusing on the features of interest.

Author Response and Action taken: We appreciate the reviewer's helpful comment on the readability of the manuscript. We decided to reorganize figures and bring back the **Supplementary Note 2** into the main **Figure 1**. We have also moved part of **Figure 1** into a new **Figure 2**. In the main text, we have added a new section named "Reassignment of the observed phonon spectra in the AFM-*a* and AFM-*b* phases" to walk the readers through the Raman data we collected. This new version better conveys the Raman analysis, selection rule consideration and provides the overview of the full spectra data both in A₁ and B₂ geometry for a reader to understand the overall temperature dependent Raman data from Ca₃Ru₂O₇.

References

- 1 Puggioni, D., Horio, M., Chang, J. & Rondinelli, J. M. Cooperative interactions govern the fermiology of the polar metal Ca₃Ru₂O₇. *Physical Review Research* **2** (2020).
- 2 Petkov, V. *et al.* Lattice distortions and the metal-insulator transition in pure and Ti-substituted Ca₃Ru₂O₇. *Journal of Physics-Condensed Matter* **35** (2023).
- 3 Yang, H. B. *et al.* Emergence of preformed Cooper pairs from the doped Mott insulating state in Bi₂Sr₂CaCu₂O_{8+delta}. *Nature* **456**, 77-80 (2008).
- 4 Baumberger, F. *et al.* Nested Fermi surface and electronic instability in Ca₃Ru₂O₇. *Phys Rev Lett* **96** (2006).
- 5 Lee, J. S. *et al.* Pseudogap dependence of the optical conductivity spectra of Ca₃Ru₂O₇: A possible contribution of the orbital flip excitation. *Phys Rev Lett* **98** (2007).
- 6 Yoshida, Y. *et al.* Crystal and magnetic structure of Ca₃Ru₂O₇. *Phys Rev B* **72** (2005).
- 7 Yan, J., Zhang, Y. B., Kim, P. & Pinczuk, A. Electric field effect tuning of electron-phonon coupling in graphene. *Phys Rev Lett* **98** (2007).
- 8 Lazzeri, M., Piscanec, S., Mauri, F., Ferrari, A. C. & Robertson, J. Phonon linewidths and electron-phonon coupling in graphite and nanotubes. *Phys Rev B* **73** (2006).
- 9 Peng, J. *et al.* From quasi-two-dimensional metal with ferromagnetic bilayers to Mott insulator with G-type antiferromagnetic order in Ca₃(Ru_{1-x}Ti_x)₂O₇. *Phys Rev B* **87** (2013).
- 10 Iliev, M. N. *et al.* Raman spectroscopy of Ca₃Ru₂O₇: Phonon line assignment and electron scattering. *Phys Rev B* **71** (2005).
- 11 Popov, M. N. *et al.* Raman spectra of fine-grained materials from first principles. *Npj Comput Mater* **6** (2020).
- 12 Wang, C. H., Jing, X. P., Feng, W. & Lu, J. Assignment of raman-active vibrational modes of MgTiO₃. *J Appl Phys* **104** (2008).
- 13 Zheng, J. Z. *et al.* Anharmonicity-induced phonon hardening and phonon transport enhancement in crystalline perovskite BaZrO₃. *Phys Rev B* **105** (2022).
- 14 Hellman, O., Steneteg, P., Abrikosov, I. A. & Simak, S. I. Temperature dependent effective potential method for accurate free energy calculations of solids. *Phys Rev B* **87** (2013).
- 15 Zhou, M. J. *et al.* First-principles lattice dynamics and thermodynamic properties of pre-perovskite PbTiO₃. *Acta Mater* **171**, 146-153 (2019).
- 16 He, L. H. *et al.* Accuracy of generalized gradient approximation functionals for density-functional perturbation theory calculations. *Phys Rev B* **89** (2014).
- 17 Yuan, Y. K. *et al.* Ultrafast quasiparticle dynamics in the correlated semimetal Ca₃Ru₂O₇. *Phys Rev B* **99** (2019).
- 18 Arpaia, R. *et al.* Dynamical charge density fluctuations pervading the phase diagram of a Cu-based high-T_c superconductor. *Science* **365**, 906-+ (2019).
- 19 Yokoya, T., Kiss, T., Chainani, A., Shin, S. & Yamaya, K. Role of charge-density-wave fluctuations on the spectral function in a metallic charge-density-wave system. *Phys Rev B* **71** (2005).
- 20 Horio, M. *et al.* Electronic reconstruction forming a C₂-symmetric Dirac semimetal in Ca₃Ru₂O₇. *Npj Quantum Mater* **6** (2021).

- 21 Loret, B. *et al.* Intimate link between charge density wave, pseudogap and superconducting energy scales in cuprates. *Nat Phys* **15**, 771-775 (2019).
- 22 Auvray, N. *et al.* Nematic fluctuations in the cuprate superconductor $\text{Bi}_2\text{Sr}_2\text{CaCu}_2\text{O}_{8+\delta}$. *Nat Commun* **10** (2019).
- 23 Benhabib, S. *et al.* Collapse of the Normal-State Pseudogap at a Lifshitz Transition in the $\text{Bi}_2\text{Sr}_2\text{CaCu}_2\text{O}_{8+\delta}$ Cuprate Superconductor. *Phys Rev Lett* **114** (2015).
- 24 Gallais, Y. *et al.* Observation of Incipient Charge Nematicity in $\text{Ba}(\text{Fe}_{1-x}\text{Co}_x)_2\text{As}_2$. *Phys Rev Lett* **111** (2013).

REVIEWERS' COMMENTS

Reviewer #1 (Remarks to the Author):

The authors satisfactorily addressed my comments and improved the manuscript. I only have one minor comment concerning the reply, which is not critical. In the reply, the authors introduced the dynamical matrix $D_{\{Ia,Jb\}}$. But the definition the authors wrote is in fact the interatomic force constant (which does not depend on momentum). The Fourier transform of the interatomic force constant is the dynamical matrix, which is momentum-dependent. Diagonalizing the dynamical matrix yields the phonon frequency.

Reviewer #3 (Remarks to the Author):

I found the revised manuscript significantly improved over the previous version. Moreover, I was positively surprised by the actions taken by the authors to address my comments. In particular, I appreciated the inclusion of the Ti doping measurement and the test of different DFT functionals. I believe that the paper is now fit for publication in Nature Communications.

Response to Reviewer Comments

NCOMMS-22-53206A: Strong Electron-Phonon Coupling driven Pseudogap Modulation and Density-Wave Fluctuations in a Correlated Polar Metal

Contents

- i. Response summary
- ii. Point-by-point response – Reviewer #1
- iii. Response to Reviewer #3

Response summary

We sincerely thank the reviewers for their positive feedbacks. We made corresponding changes in the main text to address the comment of reviewer #1. We highlight the changes made in the main text compared to last revision.

Reviewer #1 (Remarks to the Author, Author Responses, and Actions Taken):

Reviewer Comment: The authors satisfactorily addressed my comments and improved the manuscript. I only have one minor comment concerning the reply, which is not critical. In the reply, the authors introduced the dynamical matrix $D_{\{I\alpha, J\beta\}}$. But the definition the authors wrote is in fact the interatomic force constant (which does not depend on momentum). The Fourier transform of the interatomic force constant is the dynamical matrix, which is momentum-dependent. Diagonalizing the dynamical matrix yields the phonon frequency.

Author Response: We thank the reviewer for pointing out the ambiguity in our definition of dynamical matrix. Here is the revised definition of weighted atomic unit:

Definition of u : We represent phonons using mass-weighted atomic displacements $Q_{I\alpha} \equiv \sqrt{m_I} u_{I\alpha}$, where I is the atom index, α is the space direction, m_I is the mass of atom I (in atomic units of mass, a.m.u = $1.6605 \cdot 10^{-24}$ g), and $u_{I\alpha}$ is the displacement of atom I along direction α (in Å). The dynamical matrix $\mathbf{D}(\mathbf{q})$ (the mass-rescaled interatomic force constant matrix for monochromatic perturbations of atomic positions of wave vector \mathbf{q}) can then be expressed as

$$D_{I\alpha, J\beta}(\mathbf{q}) = \frac{1}{\sqrt{m_I m_J}} \sum_{\mathbf{R}} e^{i\mathbf{q} \cdot \mathbf{R}} \Phi_{I\alpha, J\beta}(\mathbf{R}),$$

where $\Phi_{I\alpha, J\beta}(\mathbf{R}) = \partial^2 E / \partial u_{I\alpha}(\mathbf{0}) \partial u_{J\beta}(\mathbf{R})$ is the (un-rescaled) interatomic force constant matrix between atoms $I\alpha$ and $J\beta$ residing in unit cells that are separated by the translation vector \mathbf{R} . Since the Raman response is related to the zone-center phonons ($\mathbf{q} \approx \mathbf{0}$), we consider the dynamical matrix at the Γ -point:

$$D_{I\alpha, J\beta}(\mathbf{0}) = \frac{1}{\sqrt{m_I m_J}} \sum_{\mathbf{R}} \frac{\partial^2 E}{\partial u_{I\alpha}(\mathbf{0}) \partial u_{J\beta}(\mathbf{R})} = \sum_{\mathbf{R}} \frac{\partial^2 E}{\partial Q_{I\alpha}(\mathbf{0}) \partial Q_{J\beta}(\mathbf{R})}.$$

By diagonalizing the dynamical matrix, one obtains the resonant frequencies ω_n and the normal modes \mathbf{e}_n :

$$\sum_{J\beta} D_{I\alpha, J\beta}(\mathbf{0}) e_{J\beta, n} = \omega_n^2 e_{I\alpha, n}.$$

Therefore, the atomic displacements can be decomposed along the normal modes as

$$u_{I\alpha} = \frac{Q_{I\alpha}}{\sqrt{m_I}} = \frac{1}{\sqrt{m_I}} \sum_n Q_n e_{I\alpha, n},$$

where $Q_n \equiv \sum_{I\alpha} Q_{I\alpha}^* e_{I\alpha, n}$ denotes the coordinate of the n^{th} normal mode [in (a.m.u)^{1/2} Å]. We updated the method section of the manuscript and the figures to clarify the definitions of $u_{I\alpha}$, $Q_{I\alpha}$, and Q_n , ensuring that notations are internally consistent and compliant with literature conventions.

Action Taken: The above definition is updated in the manuscript.

Reviewer #3 (Remarks to the Author, Author Responses, and Actions Taken):

Reviewer Comment: I found the revised manuscript significantly improved over the previous version. Moreover, I was positively surprised by the actions taken by the authors to address my comments. In particular, I appreciated the inclusion of the Ti doping measurement and the test of different DFT functionals. I believe that the paper is now fit for publication in Nature Communications.

Author Response: We are happy to hear that the reviewer finds our revised manuscript fit for publication. We appreciate their crucial feedbacks that motivated us to further clarify our phonon assignment.